# Further improvement and evaluation of nudging in the E3SM Atmosphere Model version 1 (EAMv1): simulations of the mean climate, weather events, and anthropogenic aerosol effects

Shixuan Zhang[1], Kai Zhang[1], Hui Wan[1], and Jian Sun[1,2]

[1]Pacific Northwest National Laboratory, Richland, Washington, USA
[2]National Center for Atmospheric Research, Boulder, CO, USA

**Correspondence:** Shixuan Zhang (Shixuan.Zhang@pnnl.gov) and Kai Zhang (Kai.Zhang@pnnl.gov)

**Abstract.** A previous study on the use of nudging in EAMv1 had an unresolved issue, namely a simulation nudged to EAMv1's own meteorology showed non-negligible deviations from the free-running baseline simulation over some of the subtropical marine stratocumulus and trade cumulus regions. Here, we demonstrate that the deviations can be substantially reduced by (1) changing where the nudging tendency is calculated in the time integration loop of a nudged EAM simulation so as to improve consistency with the free-running baseline and (2) increasing the frequency of the constraining data so as to better capture strong sub-diurnal variations.

The fact that modification (2) improves the climate representativeness of the nudged simulations has motivated us to investigate whether the use of newer reanalysis products with higher data frequency can help improve nudged hindcast simulations by better capturing the observed weather events. To answer this question, we present simulations conducted at EAMv1's standard horizontal resolution (approximately 1°) with nudging towards 6-hourly ERA-Interim reanalysis or 6-hourly, 3-hourly, or hourly ERA5 reanalysis. These simulations are evaluated against the climatology of free-running EAMv1 simulations as well as reanalyses, satellite retrievals, and in-situ measurements from the Atmospheric Radiation Measurement user facility. For the 1° EAMv1 simulations, we recommend using the relocated nudging tendency calculation and the ERA5 reanalysis at 3-hourly or higher frequency.

Simulations used for estimating the anthropogenic aerosol effects often use nudging to help discern signal from noise. The sensitivity of such estimates to the configuration of nudging is investigated in EAMv1, again using the standard 1° horizontal resolution. We find that when estimating the global mean effects, the frequency of constraining data has relatively small impacts, while the choice of nudged variables can change the results substantially. The nudging of air temperature (in addition to horizontal winds) has two non-negligible effects: First, when the constraining data comes from reanalysis, the nudging-induced mean bias correction can cause significant changes in the simulated clouds and hence substantially different estimates of the aerosol effects. The impact of the mean bias correct on ice cloud formation has been noted in previous studies and is also seen in EAMv1. For applications like ours, where the preferred configurations of nudging are those capable of providing results consistent with the multi-year free-running simulations, the consequence of the mean bias correction is undesirable. The second important impact of temperature nudging is a significant suppression of adjustments to aerosol forcing, which also causes changes in the estimated aerosol effects. This effect can be seen in simulations nudged to either reanalysis or EAM's

own meteorology. These results suggest that nudging horizontal winds but not temperature is a better choice for estimating the anthropogenic aerosol effects.

## 1 Introduction

Nudging (or Newtonian relaxation) is widely used for diagnosing sensitivities of climate simulations to modifications in model formulation and parameters (Lohmann and Hoose, 2009; Zhang et al., 2012; Separovic et al., 2012; Lin et al., 2016) as well as changes in computational methods (e.g., Wan et al., 2014) and external forcing (Kooperman et al., 2012; Zhang et al., 2014). It has been shown that by constraining the large-scale meteorological conditions (e.g. horizontal winds) toward weather reanalysis or a baseline simulation, nudging can help reduce noise caused by natural variability and hence allow for the detection of signals without long simulations or large ensembles (e.g., Kooperman et al., 2012). However, nudging should be used with care. The configuration of the nudged simulations must be carefully evaluated based on the purpose of the sensitivity experiment. Many studies have shown that the forcing terms introduced by nudging can be sufficiently strong to break the internal balance between the resolved dynamics and parameterized physics (e.g. Jeuken et al., 1996) or to cause significant changes in the model's climate (e.g. Zhang et al., 2014), making the results less useful for interpreting the behavior of the original model.

Sun et al. (2019) evaluated two types of nudged simulations conducted with the atmosphere component of the Energy Exascale Earth System Model version 1 (EAMv1, Rasch et al., 2019; Xie et al., 2018) at the standard hoirzontal resolution with approximately $1^o$ grid spacing. One type of the simulations was constrained by reanalysis products and the second type was constrained by meteorological fields written out from a free-running baseline simulation conducted with the same model (hereafter referred to as the "baseline nudging" method). They showed that simulations using baseline nudging closely resembled the free-running simulation for the key meteorological variables evaluated therein, as evidenced by the high spatial and temporal correlations between the nudged and free-running simulations. On the other hand, systematic decreases in the annual mean shortwave cloud radiative forcing (SWCF) were seen in subtropical and tropical regions when nudging was used, with local annual averages as large as $8 \text{ W m}^{-2}$. The discrepancies are inconvenient as they result in inaccuracies in the anthropogenic aerosol effects estimated using baseline nudging.

The study presented here starts with an effort to address these discrepancies. The sequence of calculations related to nudging in EAMv1's time integration loop is reviewed (Sections 2.2 and 3.1) and the time-step-by-time-step temporal evolution of the model state in the subtropics is analyzed (Section 3.2). We demonstrate that the discrepancy issue in $1^o$ simulations in Sun et al. (2019) can be substantially alleviated by two revisions of the nudging implementation: first, changing the sequence of calculations in a nudged EAM simulation to improve consistency with the free-running baseline; second, increasing the frequency of constraining data from 6-hourly to 3-hourly to better capture strong sub-diurnal variations. The resulting improvements in climate representativeness are presented in Section 3.

Motivated by the improvements, additional simulations and analyses are presented in Section 4 to explore the potential benefits of using newer reanalysis products with higher data frequency in nudged simulations that aim at capturing the observed weather events. In many previous studies (e.g., Telford et al., 2008; Zhang et al., 2014), the reanalysis products used for

generating the nudging data were available only 4 times per day. This was the case, for example, for the ERA-Interim reanalysis (Dee et al., 2011) from the European Centre for Medium-Range Weather Forecasts (ECMWF) as well as the reanalysis of Kanamitsu et al. (2002) from the National Centers for Environmental Prediction and National Center for Atmospheric Research (NCEP/NCAR). In recent years, reanalysis data with higher temporal frequency are emerging. For example, MERRA-2 (Gelaro et al., 2017) from the National Aeronautics and Space Administration's Global Modeling and Assimilation Office is available every 3 hours, while the ERA5 reanalysis from ECMWF (Hersbach et al., 2020) has hourly data. On the one hand, using high-frequency reanalysis data for nudging may better constrain a simulation. On the other hand, processing more data before and during a simulation will consume more resources for data processing and storage. Therefore, it is useful to evaluate the benefit of using high-frequency nudging data. Furthermore, since ERA5 is a new reanalysis product that has not been widely used for nudged simulations, it is useful to compare simulations nudged towards ERA5 and ERA-interim, evaluate hindcast skills of these simulations, and provide a recommendation. For those purposes, we present in Section 4 simulations constrained using 6-hourly ERA-Interim reanalysis and 6-hourly, 3-hourly, or hourly ERA5 reanalysis. Hindcast skills of the nudged $1^o$ EAMv1 simulations are evaluated against global-scale satellite retrievals of outgoing longwave radiation and precipitation, as well as in-situ measurements of air temperature, humidity, and horizontal winds from the Atmospheric Radiation Measurement (ARM) user facility. Since one of our primary interests in using nudged simulations is to efficiently estimate the climate impacts of anthropogenic aerosols, we present in Section 5 some analysis of the sensitivity of the estimate to the configuration of nudged simulations. Our findings and recommendations are summarized in Section 6.

## 2 Model and simulations

### 2.1 A brief overview of EAMv1

E3SM is a global Earth system model developed by the U.S. Department of Energy (Golaz et al., 2019). The present study focuses on nudging applications in the E3SM Atmosphere Model version 1 (EAMv1; Rasch et al., 2019; Xie et al., 2018). EAMv1 uses the hydrostatic spectral element (SE) dynamical core on a cubed-sphere mesh (Dennis et al., 2012; Taylor et al., 2010) to solve the equations for large-scale dynamics and tracer transport. The key subgrid-scale physical processes considered in EAMv1 include deep convection (hereafter Deep Cu; Zhang and McFarlane, 1995), turbulence and shallow convection (Golaz et al., 2002; Larson et al., 2002), cloud microphysics (Morrison and Gettelman, 2008; Gettelman and Morrison, 2015; Wang et al., 2014), aerosol life cycle (Liu et al., 2016; Wang et al., 2020), and radiation (Iacono et al., 2008; Mlawer et al., 1997). EAMv1 is interactively coupled with a land model (Oleson et al., 2013).

Figure 1a shows the sequence of dynamics and physics calculations (i.e., the time integration loop) in EAMv1. More detailed descriptions of the time stepping and coupling of physics and dynamics can be found in Zhang et al. (2018) and Wan et al. (2021, 2022). One important feature relevant to the discussion below is that most of the atmospheric processes are numerically coupled using sequential splitting. This means after a model component (e.g., a parameterization) predicts the rate-of-change (also called tendency) of the model state caused by the atmospheric process it represents, the model state will be updated using the predicted tendency before being handled to the next model component (e.g., another parameterization).

The simulations presented in this paper use a horizontal resolution of approximately 1° (∼110 km). There are 72 layers in the vertical, extending from the Earth's surface to ∼0.1 hPa (∼64 km). The vertical grid spacing is uneven, with the layer thickness ranging typically from 20 m to 100 m near the surface and up to 600 m near the model top.

## 2.2 Nudging in EAMv1

The nudging implementation in EAMv1 was described and evaluated in Sun et al. (2019), so we only provide a brief introduction here. Nudging constrains the model solution toward prescribed atmospheric conditions for a certain variable by adding a relaxation term to the prognostic equation:

$$\left(\frac{\partial X_m}{\partial t}\right)_{\text{ndg}} = -\frac{X_m - X_p}{\tau}, \tag{1}$$

where $X$ in Eq. (1) represents a model state variable like horizontal winds (U, V), temperature (T), or specific humidity (Q). Subscript $m$ refers to the model-predicted value. Subscript $p$ indicates the prescribed field that is taken or derived from either a global weather reanalysis or a free-running simulation using the same model. $\tau$ denotes the relaxation time scale. All three quantities, $X_m$, $X_p$, and $\tau$, can affect the sign and strength of the nudging-induced forcing.

Pink boxes in the left panel of Figure 1 illustrate where the nudging-related calculations occur in the default EAMv1. In a nudged simulation, after the resolved dynamics (see blue box in figure) has been calculated, a nudging tendency term in the form of Eq. (1) is calculated for each nudged variable with $X_m$ being the value of $X$ after the dynamical core. After the entire physics parameterization suite has been calculated, the sum of the parameterization-induced tendencies and the nudging tendencies are passed to the physics-dynamics coupling interface.

It is worth noting that, when an EAM simulation is considered to be a baseline simulation, the dynamical and thermodynamical variables (e.g., U, V, T, Q, and the surface pressure PS) that are archived – and subsequently used in a nudged simulation as the prescribed atmospheric state – are the values saved before the radiation calculation (see pink dashed box in Fig. 1a). In other words, in the default EAMv1, the $X_p$ in the right-hand side of Eq. (1) is archived before radiation while the $X_m$ in that same equation corresponds to the model state after the dynamical core. As is discussed in Section 3.1, the fact that $X_p$ and $X_m$ correspond to different locations in the time integration loop plays an important role in causing the issue in Sun et al. (2019) that motivated this study.

## 2.3 Simulations

The EAMv1 simulations presented in this paper are summarized in Table 1. All the simulations involved active atmosphere and land but used prescribed sea surface temperature (SST) and sea ice extension, following the protocol from the Atmospheric Model Intercomparison Project (Gates et al., 1999). The SST and sea-ice extension used in this study are weekly data from the National Oceanic and Atmospheric Administration (NOAA) Optimum Interpolation (OI) analysis (Reynolds et al., 2002). Other external forcings, including volcanic aerosols, solar variability, concentrations of greenhouse gases, and anthropogenic emissions of aerosols and their precursors, were prescribed following the World Climate Research Programme (WCRP) Coupled Model Intercomparison Project Phase 6 (CMIP6; Eyring et al., 2016; Hoesly et al., 2018; Feng et al., 2020).

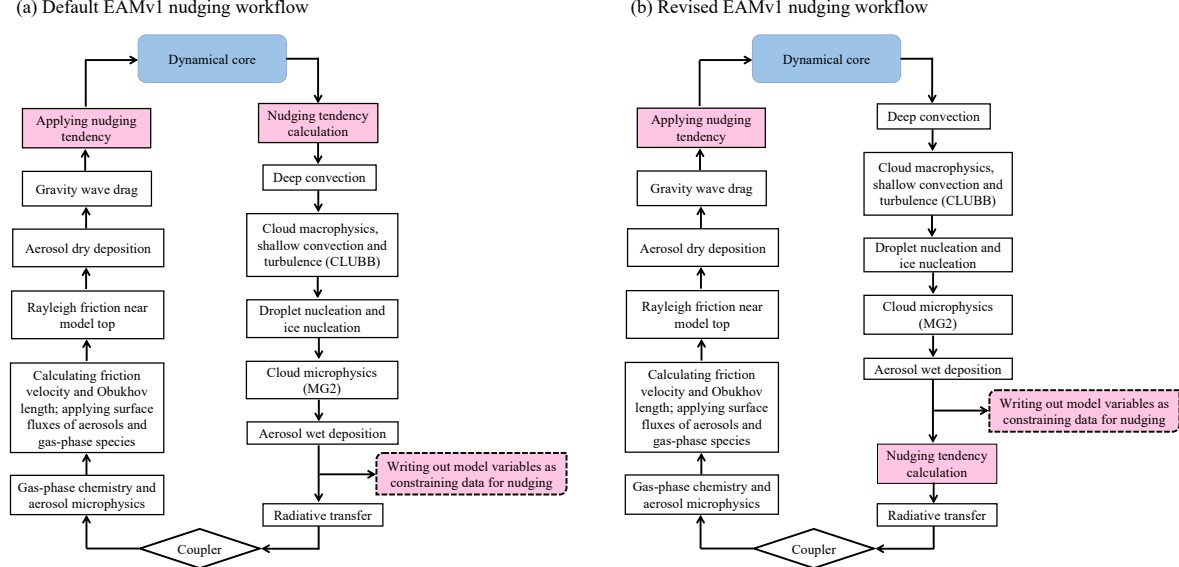

**Figure 1.** Flowcharts showing the sequence of dynamics and physics calculations within one time step in an EAMv1 simulation. Pink boxes indicate where the nudging-related calculations occur. Panel (a) is adapted from Fig. S1 in Sun et al. (2019) and corresponds to the default EAMv1 code. Panel (b) is the revised sequence of calculations we recommend. The key difference is that in panel (b), the calculation of nudging tendency using Eq. (1) occurs at the same location where the meteorological fields are written out from the baseline simulation, i.e., before the radiation parameterization. Panel (b) is described in detail in Section 3.1.

All simulations were performed from 1 October 2009 to 31 December 2010. The first 3 months were discarded as model spin-up, and the remaining 1 year of model output was used for analysis. The choice of simulation year was based on convenience, as hourly ERA5 data of 2010 were readily available to us. Sun et al. (2019) have shown that the annual mean cloud radiative forcing and its shortwave and longwave components derived from 1-year nudged simulations are representative of the corresponding longer-term (e.g., 5-year) statistics (see, e.g., Fig. 19 therein).

The anthropogenic aerosol effects we are interested in estimating in this study is the effective radiative forcing (ERF) defined in the Fifth Assessment Report of the Intergovernmental Panel on Climate Change, namely the changes in the TOA radiative fluxes when all physical variables in a climate model are allowed to respond to perturbations except for those concerning the ocean and sea ice (Myhre et al., 2014). Our primary focus is the net TOA flux and its shortwave and longwave components. These are denoted by $F_{NET}$, $F_{SW}$ and $F_{LW}$, respectively, in the remainder of the paper, with positive values indicating fluxes downward (i.e., into the atmosphere). For the readers who have worked with EAM's output, we note that the $F_{SW}$ presented here is EAM's output variable FSNT while the $F_{LW}$ here is $-$FLNT, as FLNT in EAM is defined to be positive upward. The net flux is calculated as

$$F_{NET} = F_{SW} + F_{LW} = FSNT - FLNT$$

The changes in $F_{NET}$, $F_{SW}$ and $F_{LW}$ caused by anthropogenic aerosols are denoted by $\Delta F_{NET}$, $\Delta F_{SW}$ and $\Delta F_{LW}$, respectively.

We are also interested in the impact of anthropogenic aerosols on the cloud radiative effect (CRE). CRE is defined as the change in a TOA radiative flux caused by the presence of clouds; here we denote the CRE on the net, shortwave and longwave TOA radiative flux by $CRE_{NET}$, $CRE_{SW}$ and $CRE_{LW}$, respectively, with positive values indicating more fluxes into the atmosphere. The $CRE_{SW}$ and $CRE_{LW}$ presented here are EAM's output variables SWCF and LWCF, respectively, both of which are diagnosed during a simulation by performing the radiation calculations twice (with and without clouds) and then computing the difference. The net CRE is calculated by

$$CRE_{NET} = CRE_{SW} + CRE_{LW} = SWCF + LWCF$$

The changes in $CRE_{NET}$, $CRE_{SW}$ and $CRE_{LW}$ caused by anthropogenic aerosols are denoted by $\Delta CRE_{NET}$, $\Delta CRE_{SW}$ and $\Delta CRE_{LW}$, respectively.

To estimate the anthropogenic aerosol effects (i.e., the $\Delta$ quantities) mentioned above, pairs of simulations were conducted. Each pair had identical experimental setup except that the emissions of aerosols and their precursor gases were set to the values of the year 2010 to represent the present-day (PD) condition in one simulation and the values of the year 1850 to represent the pre-industrial (PI) condition in the second simulation. The greenhouse gas concentrations, SST, and sea ice extent were unchanged (i.e., fixed at their year-2010 values). The main differences between PI and PD aerosol emissions included anthropogenic emissions of sulfate, black carbon, organic carbon, primary organic carbon, and the precursors of secondary organic aerosols (applied as yields). Biomass burning emissions were also different between the PD and PI conditions. Dust, sea salt, and marine organic aerosol emissions were calculated online using the surface wind speed and surface properties predicted in each simulation.

Three groups of simulations are presented in this paper. The first group consists of five pairs of 15-month simulations. The first pair is two free-running baseline simulations referred to as CLIM PD and CLIM PI in the remainder of the paper. From the CLIM PD simulation, the before-radiation values of U, V, T, Q, and PS were archived at 1-hour, 3-hour, and 6-hour frequencies to constrain some of the subsequent simulations. The other four pairs in group one were nudged to 6-hourly temperature output from the CLIM PD simulation but using long relaxation time scales of 10 days, 10.1 days, 10.2 days, and 10.3 days, respectively. These relaxation time scales correspond to values of $1/\tau$ on the order of $10^{-6}$, which resulted in physically insignificant constraints on the simulations. Therefore, the four pairs of nudged simulations can effectively be considered to be free-running although with perturbations introduced to the 3D temperature field that can be used to quantify natural variability in the evolution of the atmospheric state. A similar experimentation strategy has been used by Liu et al. (2018) to generate hindcast ensembles to investigate the radiative forcing of fire-emitted aerosols.

The second group of simulations was nudged to the meteorology archived from the CLIM PD simulation in group 1, regardless of whether the PD or PI emissions were used in the nudged simulations. Nudging was applied at every time step and vertical level using a 6 h relaxation time scale. The simulations labeled DNDG_UV6 and DNDG_UVT6 used the sequence of calculations shown in Fig. 1a (i.e., the default EAMv1) while RNDG_UV6 and RNDG_UVT6 used the revised sequence

shown in Fig. 1b and explained in Section 3.1. The impact of the revised sequence of calculations is evaluated in Section 3.1. The difference between the experiments labeled with "_UV" and "_UVT" is whether only the horizontal winds were nudged ("_UV") or both winds and temperature were nudged ("_UVT"). The ending number 6 in a experiment name indicates the use of 6-hourly output from CLIM PD. Additional simulations were conducted, also using the revised sequence of calculations but constrained by 3-hourly or 1-hourly output from the CLIM PD simulation (RNDG_UV3 and RNDG_UVT3; RNDG_UV1 and RNDG_UVT1). These simulations are compared with RNDG_UV6 and RNDG_UVT6 in Section 3.2 to evaluate the impact of the frequency of the constraining data .

The third group of simulations was nudged toward two reanalysis products, ERA-Interim (Dee et al., 2011) and ERA5 (Hersbach et al., 2020), to assess whether using a newer product (ERA5) and its higher data frequency, instead of the older ERA-Interim at 6-hour intervals, can provide nudged hindcast simulations that agree better with the observational data. The reanalysis products were spatially remapped to the cubed-sphere grid and 72 model layers used by EAMv1, following the method used in the Community Earth System Model Version 2 (CESM2; https://ncar.github.io/CAM/doc/build/html/users_guide/physics-modifications-via-the-namelist.html#nudging). Topographical differences between EAMv1 and the reanalysis model were taken into account during the vertical interpolation. This group of simulations are compared to global or quasi-global observational data of surface precipitation rate and the top-of-the-atmosphere (TOA) outgoing longwave radiation (OLR) from satellite retrievals (Section 4.2) as well as in-situ measurements from the Atmospheric Radiation Measurement (ARM) user facility (Section 4.3).

## 3 Improving the climate representativeness of simulations nudged to CLIM

This section focuses on analyzing the PD simulations listed in group 2 of Table 1. The CLIM PD simulation in group 1 is used as the baseline simulation and referred to as CLIM for brevity.

Before this work, the EAMv1 simulations nudged to 6-hourly output from CLIM were known to show non-negligible differences from CLIM. For example, Fig. 15b in Sun et al. (2019) showed the weakening of 1-year mean SWCF ($= \mathrm{CRF_{SW}}$ here) on the order of 2–8 W m$^{-2}$ in large areas of the subtropical marine and coastal regions when horizontal winds and temperature were both nudged. The differences exceeded 8 W m$^{-2}$ in some regions over the southeast Pacific Ocean and South America. Figure 15a in that same paper showed that constraining only the horizontal winds (i.e., no temperature nudging) would remove the discrepancies in most of the subtropical regions, although one would find 4–8 W m$^{-2}$ of strengthening of the annual mean SWCF close to the coast of Peru. The corresponding discrepancies seen in $\mathrm{CRE_{NET}}$ and cloud cover are shown in panels (b) and (d) of Fig. 2 and Fig. C1 in this paper. When winds and temperature are both nudged, we see a substantial number of grid cells in the subtropical Pacific and Atlantic oceans where the relative differences on the order of 10% to 20% are seen in $\mathrm{CRE_{NET}}$ when compared to the annual mean $\mathrm{CRE_{NET}}$ in the baseline simulation CLIM (Fig. C2d). Discrepancies of such magnitudes are counterintuitive since the constraining data were generated from the same model driven by the same external forcing. On the other hand, since nudging introduces forcing terms in the form of Eq. (1) to the model's governing equations, any differences between $X_m$ and $X_p$ will lead to deviations from a free-running simulation. Below, we

**Table 1.** List of simulations analyzed in this study. Nudging was applied at each physics time step. The use of present-day (PD, year 2010) or pre-industrial (PI, year 1850) emissions of anthropogenic aerosols and precursors is indicated in the rightmost column. Other than these emissions, all external forcing was set to the PD values.

| Group number | Simulation short name | Flowchart | Nudged variables | Constraining data and frequency | Nudging relaxation time scale | Aerosol and precursor gas emissions |
|---|---|---|---|---|---|---|
| 1 | CLIM | Fig. 1a | None | N/A | N/A | PD and PI |
| 1 | CLIMp1 | Fig. 1a | T | CLIM PD (6 hr) | 10.1 days | PD and PI |
| 1 | CLIMp2 | Fig. 1a | T | CLIM PD (6 hr) | 10.2 days | PD and PI |
| 1 | CLIMp3 | Fig. 1a | T | CLIM PD (6 hr) | 10.3 days | PD and PI |
| 1 | CLIMp4 | Fig. 1a | T | CLIM PD (6 hr) | 10.4 days | PD and PI |
| 2 | DNDG_UV6 | Fig. 1a | U, V. | CLIM PD (6 hr) | 6 hr | PD |
| 2 | DNDG_UVT6 | Fig. 1a | U, V, T | CLIM PD (6 hr) | 6 hr | PD and PI |
| 2 | RNDG_UV6 | Fig. 1b | U, V | CLIM PD (6 hr) | 6 hr | PD and PI |
| 2 | RNDG_UVT6 | Fig. 1b | U, V, T | CLIM PD (6 hr) | 6 hr | PD and PI |
| 2 | RNDG_UV3 | Fig. 1b | U, V | CLIM PD (3 hr) | 6 hr | PD and PI |
| 2 | RNDG_UVT3 | Fig. 1b | U, V,T | CLIM PD (3 hr) | 6 hr | PD and PI |
| 2 | RNDG_UV1 | Fig. 1b | U, V | CLIM PD (1 hr) | 6 hr | PD |
| 2 | RNDG_UVT1 | Fig. 1b | U, V, T | CLIM PD (1 hr) | 6 hr | PD |
| 3 | DNDG_ERAI_UV6 | Fig. 1a | U, V | ERA-Interim (6 hr) | 6 hr | PD |
| 3 | DNDG_ERAI_UVT6 | Fig. 1a | U, V, T | ERA-Interim (6 hr) | 6 hr | PD |
| 3 | RNDG_ERAI_UV6 | Fig. 1b | U, V | ERA-Interim (6 hr) | 6 hr | PD |
| 3 | RNDG_ERAI_UVT6 | Fig. 1b | U, V, T | ERA-Interim (6 hr) | 6 hr | PD |
| 3 | RNDG_ERA5_UV6 | Fig. 1b | U, V | ERA5 (6 hr) | 6 hr | PD and PI |
| 3 | RNDG_ERA5_UV3 | Fig. 1b | U, V | ERA5 (3 hr) | 6 hr | PD and PI |
| 3 | RNDG_ERA5_UVT6 | Fig. 1b | U, V, T | ERA5 (6 hr) | 6 hr | PD and PI |
| 3 | RNDG_ERA5_UVT3 | Fig. 1b | U, V, T | ERA5 (3 hr) | 6 hr | PD and PI |
| 3 | RNDG_ERA5_UVT1 | Fig. 1b | U, V, T | ERA5 (1 hr) | 6 hr | PD |

show that such deviations can be significantly reduced by revising the sequence of calculations in nudged simulations and thereby achieving better consistency with the free-running baseline (Section 3.1), as well as by increasing the data frequency of the constraining meteorology to better capture higher-frequency variations in time (Section 3.2).

### 3.1 Calculation of the nudging tendency

As mentioned in Section 2.2, in EAMv1's nudging implementation before this study, the baseline simulation's atmospheric state was archived before the radiation parameterization while the nudging-induced forcing (i.e., Eq. 1) was calculated after the dynamical core. Since EAMv1 uses sequential splitting to couple most of the atmospheric processes (see Section 2.1), if we use a subscript "DYN" to label the atmospheric state after the dynamical core and a subscript "ARC" to label the atmospheric

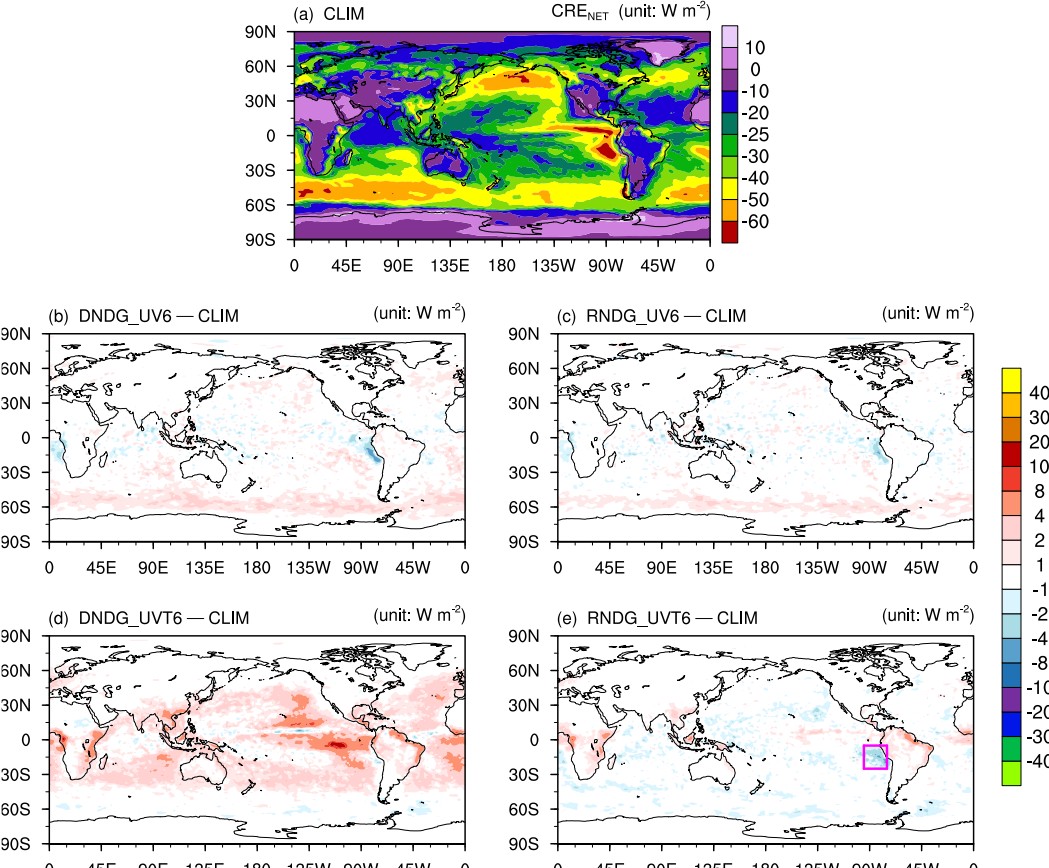

**Figure 2.** Annual mean $\mathrm{CRE_{NET}}$ (unit: W m$^{-2}$) in the free-running simulation (CLIM, panel a) and the differences between nudged simulations and CLIM (panels b-e). All simulations in this figure used the PD aerosol and precursor emissions. The nudged simulations were constrained by EAMv1s own meteorology. Descriptions of the simulation setups can be found in Section 2.3 and Table 1. The magenta box over the Peruvian stratocumulus region in panel (e) is further analyzed in Fig. 3.

state being archived, then the the old nudging implementation was, effectively,

$$\left(\frac{\partial X_m}{\partial t}\right)_{\mathrm{ndg}} = -\frac{X_{m,\mathrm{DYN}} - X_{p,\mathrm{ARC}}}{\tau} \tag{2}$$

$$= \left(-\frac{X_{m,\mathrm{ARC}} - X_{p,\mathrm{ARC}}}{\tau}\right) + \left(\frac{X_{m,\mathrm{ARC}} - X_{m,\mathrm{DYN}}}{\tau}\right) \tag{3}$$

In our understanding, the first term on the right-hand side of Eq. (3) is the intended nudging tendency while the second term is inadvertent. Furthermore, the second term can be understood as the total tendency caused by deep convection, turbulence, and stratiform cloud parameterizations scaled by a factor of $\Delta t/\tau$ where $\Delta t$ is the physics time step. Since these moist processes are known to strongly affect the atmospheric state, especially temperature and humidity, it is not surprising that nudged simulations using Eq. (2) deviate from their free-running baseline.

When the calculation of the nudging tendency is moved before the radiation parameterization so that $X_p$ from the baseline simulation and $X_m$ in the nudged simulation come from the same location of the time integration cycle (see schematic in Fig. 1b), we have, as intended,

$$\left(\frac{\partial X_m}{\partial t}\right)_{\mathrm{ndg}} = -\frac{X_{m,\mathrm{ARC}} - X_{p,\mathrm{ARC}}}{\tau}. \tag{4}$$

Sensitivity experiments confirm that using Eq. (4) instead of Eq. (2) significantly reduces discrepancies between the UVT-
210 nudged and free-running simulations, as can be seen by comparing Fig. 2e with 2d. The annual mean $\mathrm{CRE}_{\mathrm{NET}}$ differences are reduced to within 1 W m$^{-2}$ for the majority of the grid cells and within 2 W m$^{-2}$ in the subtropics and tropics, with only a small number of grid cells showing differences between 2–5 W m$^{-2}$. The discrepancies between UV-nudged and free-running simulations are also reduced, although not as significantly (Fig. 2c versus 2b). The remaining discrepancies are investigated in the next subsection.

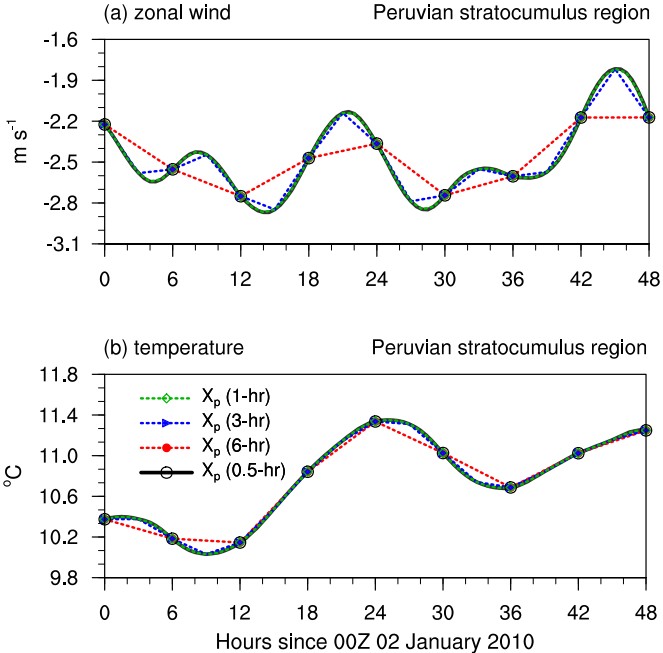

**Figure 3.** Time evaluation of (a) zonal wind (unit: m s$^{-1}$) and (b) temperature (unit: °C) at the model level closest to 700 hPa during a 48-h period starting from 00Z 02 January 2010. The values shown are horizontal averages over the magenta box in Fig. 2e. The black thick lines are time-step-by-time-step output from CLIM. The red, blue, and green lines are time-step-by-time-step values of $X_p$ in Eq. (1) that were obtained by linear temporal interpolation using 6-hourly, 3-hourly and hourly output from CLIM.

## 3.2 Frequency of the constraining data

Nudged simulations in the literature (e.g., Kooperman et al., 2012; Subramanian and Zhang, 2014; Ma et al., 2014, 2015; Lin et al., 2016; Fast et al., 2016), including our own work (e.g., Zhang et al., 2014; Sun et al., 2019), often used 6-hourly

constraining data. The historical reason was that reanalysis data used to be available only 4 times per day. Such a frequency, on the other hand, can be insufficient for capturing fast variations because of the problem of aliasing.

Figure 3 shows the evolution of lower-troposphere (700 hPa) zonal wind and temperature averaged over the Peruvian stratocumulus region marked by the magenta box in Fig. 2e, for a 2-day period starting from 00Z 02 in January 2010. In Fig. 3, the black solid lines are time-step-by-time-step output from CLIM where $\Delta t$ = 30 min. The dashed lines are the linearly interpolated time series used in the calculation of nudging tendencies; green, blue, and red correspond to cases in which the constraining data was provided at 1 h, 3 h, and 6 h frequencies, respectively. The EAMv1-simulated wind field in the Peruvian

stratocumulus region shows prominent 12 h cycles. Linear interpolation of 6-hourly data misses all the local maxima and minima (red line in Fig. 3a) while the interpolation from 3-hourly data provides substantial improvements (blue line in Fig. 3a). The temperature time series in Fig. 3b also shows 12 h variations although the amplitude is much smaller compared to the diurnal cycle.

Considering the multiscale nature of the atmospheric motions, one can speculate there are modes of variability that need

higher than 3-hourly sampling frequency to avoid aliasing. The sensitivity experiments conducted using 6 h, 3 h, and 1 h constraining data (see group 2 of Table 1 and Fig. 4), however, suggest that nudged simulations using 3-hourly data can provide annual mean cloud forcing estimates that agree with CLIM within 1 W m$^{-2}$ for most grid cells, at least for the 1° simulations considered here. In the future, before nudged simulations are conducted at substantially higher resolutions (e.g., 0.25° or convection-permitting), it will be useful to find out whether the better-resolved fine-scale motions will require higher

frequencies of constraining data.

### 3.3    Climate representativeness beyond cloud radiative forcing

The investigations discussed in Sections 3.1 and 3.2 focused on cloud radiative forcing. In Fig. 5, we further evaluate the climate representativeness of the nudged simulations by assessing the annual averages of twenty 2D fields that are often examined during model development and tuning. These fields are labeled along the x-axis in panel Fig. 5d and explained in

Appendix B. For each of the nudged PD simulations listed in group 2 of Table 1 and each of the twenty fields, we calculated two error metrics with respect to the CLIM PD simulation: one measuring the difference in the global annual mean (Fig. 5a-b) and one measuring the root-mean-square difference in the annually-averaged global geographical pattern (Fig. 5c-d).

Consistent with the cloud forcing results shown in Figs. 2 and 4, the revised sequence of calculations and 3 h data frequency have larger impacts on the UVT-nudged simulations than on UV-nudged simulations. Nevertheless, we see a systematic reduc-

tion of global mean and pattern errors across all twenty quantities evaluated in Fig. 5 (i.e., yellow bars are substantially shorter than orange bars; green bars are significantly shorter than yellow bars). In simulations RNDG_UVT3 and RNDG_UV3, the errors in global averages are reduced to less than 1% (green bars in Fig. 5a-b). The errors in geographical patterns are reduced to 2% or less for the UVT-nudged simulation and 3% or less for the UV-nudged simulation (green bars in Fig. 5c-d). Comparing panels c and d in Fig. 5, we see lower errors associated with UVT-nudging; this is possibly an indication of better

consistency between winds and temperature when both are nudged. Further increase of data frequency to 1 h only leads to limited improvements in the simulated geographical patterns. We consistently see the fact that increasing data frequency from

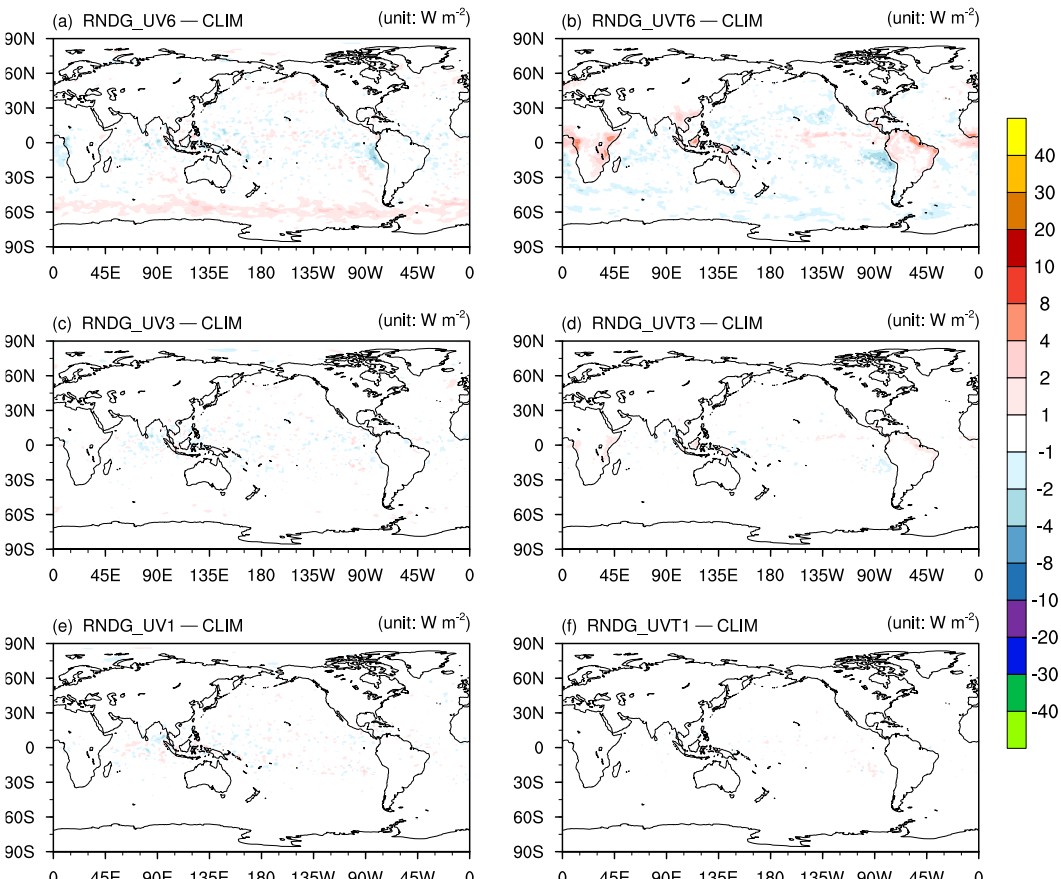

**Figure 4.** Differences in annual mean $CRE_{NET}$ (unit: W m$^{-2}$) between nudged simulations and CLIM, all using PD (year 2010) forcing conditions. Simulations shown in the left column used only wind nudging while simulations shown in the right column used wind and temperature nudging. From the first row to the bottom row, the frequency of constraining data used in the nudged simulations were 6-hourly, 3-hourly, and hourly, respectively. The simulation setups are described in Section 2.3 and Table 1.

6-hourly to 3-hourly leads to a better agreement of global averages with the free-running simulation, but a further increase to hourly data no longer leads to substantial differences. This can be seen not only in Fig. 5a-b but also in the additional cloud- and precipitation-related quantities shown in Table S2.

Therefore, for future applications that use 1 ° simulations nudged to the model's own meteorology, we recommend using the revised sequence of calculations depicted in Fig. 1b and 3-hourly constraining data. Future investigations are needed to find out whether nudged simulations at higher spatial resolutions will require more frequent constraining data.

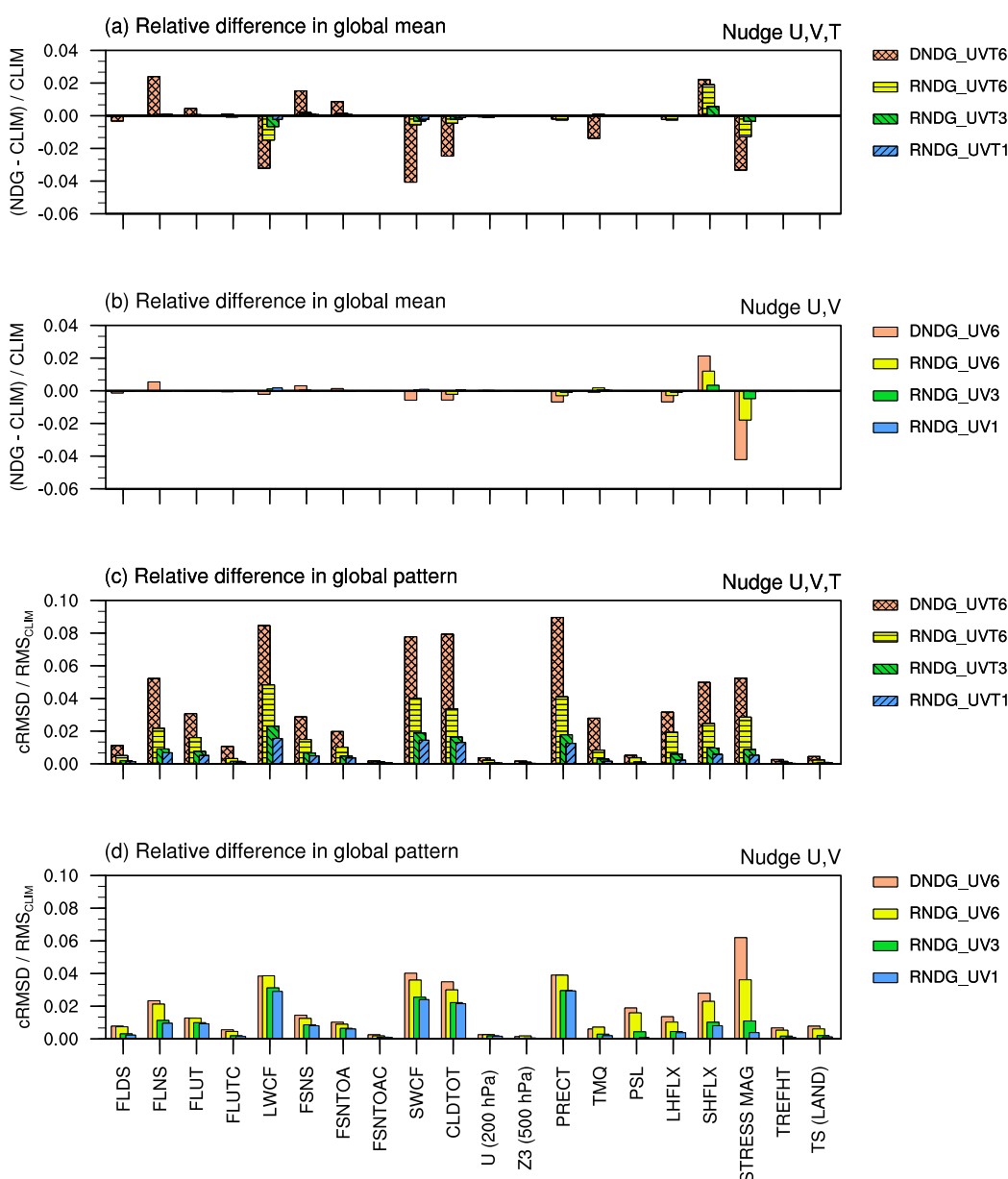

**Figure 5.** Comparison of annual averages between the nudged simulations and CLIM, all using PD (year 2010) forcing conditions. The physical quantities labeled along the x-axis are explained in Table B1 in the appendix. Panels (a) and (b) show relative differences in the simulated global averages. Panels (c) and (d) show relative differences in the simulated geographical distributions. The hatched bars correspond to UVT nudging; the bars without hatching correspond to wind-only nudging. Different colors in the same panel indicate different nudging configurations in terms of the sequence of calculations and frequency of constraining data. All differences were calculated against CLIM. Further details can be found in Section 3.3 and Appendix B.

## 4 Evaluation of the simulations nudged to reanalyses

As mentioned in the introduction, a common application of nudging is to force the simulated large-scale meteorological conditions to follow the trajectory of the observed evolution so as to facilitate process-level model evaluation or composite analyses focused on specific types of weather events. In this case, nudged simulations are typically performed using gridded reanalysis products from an operational weather prediction center as the constraining data. The findings from the previous section, especially the conclusion that higher frequency of the constraining data might help better capture important modes of variability, motivated us to evaluate the potential benefits of using more recent reanalysis products such as ERA5 (Hersbach et al., 2020) and MERRA2 (Gelaro et al., 2017). Since ERA5 has the highest data frequency (i.e., hourly), and ERA5 is also known to show better agreement with observations when compared with its predecessor ERA-Interim (Hersbach et al., 2020), we focus on ERA5-constrained simulations in this section and use the sensitivity experiments listed in group 3 of Table 1 to answer the following questions:

- What is the impact of nudging on the simulated mean climate? (Section 4.1)

- Do ERA5-nudged hindcast simulations agree better with observations than the ERA-Interim-nudged simulations? (Section 4.2)

- How frequently should the nudging data be provided to obtain sufficiently good hindcast skill? (Section 4.3)

The discussion in this section focuses on simulations performed under PD forcing conditions.

### 4.1 Global and regional mean climate

Since the long-term climate simulated by the free-running EAMv1 is known to have non-negligible biases with respect to observational data (Rasch et al., 2019; Xie et al., 2018), nudging towards reanalysis is expected to result in significant changes in the statistical features of the simulated climate. When U and V are nudged to 6-hourly meteorology from ERA-Interim, the annual mean net $\mathrm{CRE_{NET}}$ can deviate from CLIM by more than -20 W m$^{-2}$ in the Californian, Peruvian, and Namibian stratocumulus regions (Fig. 6a). When T is also nudged, we see deviations on the order of -10 W m$^{-2}$ to -20 W m$^{-2}$ over the storm tracks and 10 W m$^{-2}$ to 40 W m$^{-2}$ over the trade cumulus regions (Fig. 6b). In terms of global averages, nudging only U and V to 6-hourly ERA-Interim data gives a $\mathrm{CRE_{NET}}$ very close to the value in CLIM; the shortwave and longwave components deviate from the corresponding values in CLIM by about 0.3 W m$^{-2}$ (see simulation RNDG_ERAI_UV6 in Table S3). If T is nudged in addition to U and V, the global mean $\mathrm{CRE_{NET}}$ deviates from the value in CLIM by about -1.7 W m$^{-2}$, attributable mainly to the longwave component (see simulation RNDG_ERAI_UVT6 in Table S3). These results are consistent with the conclusion from Sun et al. (2019) that ERA-nudged runs differ substantially from CLIM.

The second row of Fig. 6 shows the impact of using ERA5 instead of ERA-Interim while keeping a 6-hourly data frequency. The resulting changes are substantially smaller than the differences between ERA-nudged simulations and CLIM, although we still see some $\mathrm{CRE_{NET}}$ differences in the subtropics and tropics as large as 10 W m$^{-2}$ to 20 W m$^{-2}$. The relatively small impact of replacing ERA-Interim with ERA5 is expected, as the differences between ERA5 and ERA-Interim are substantially

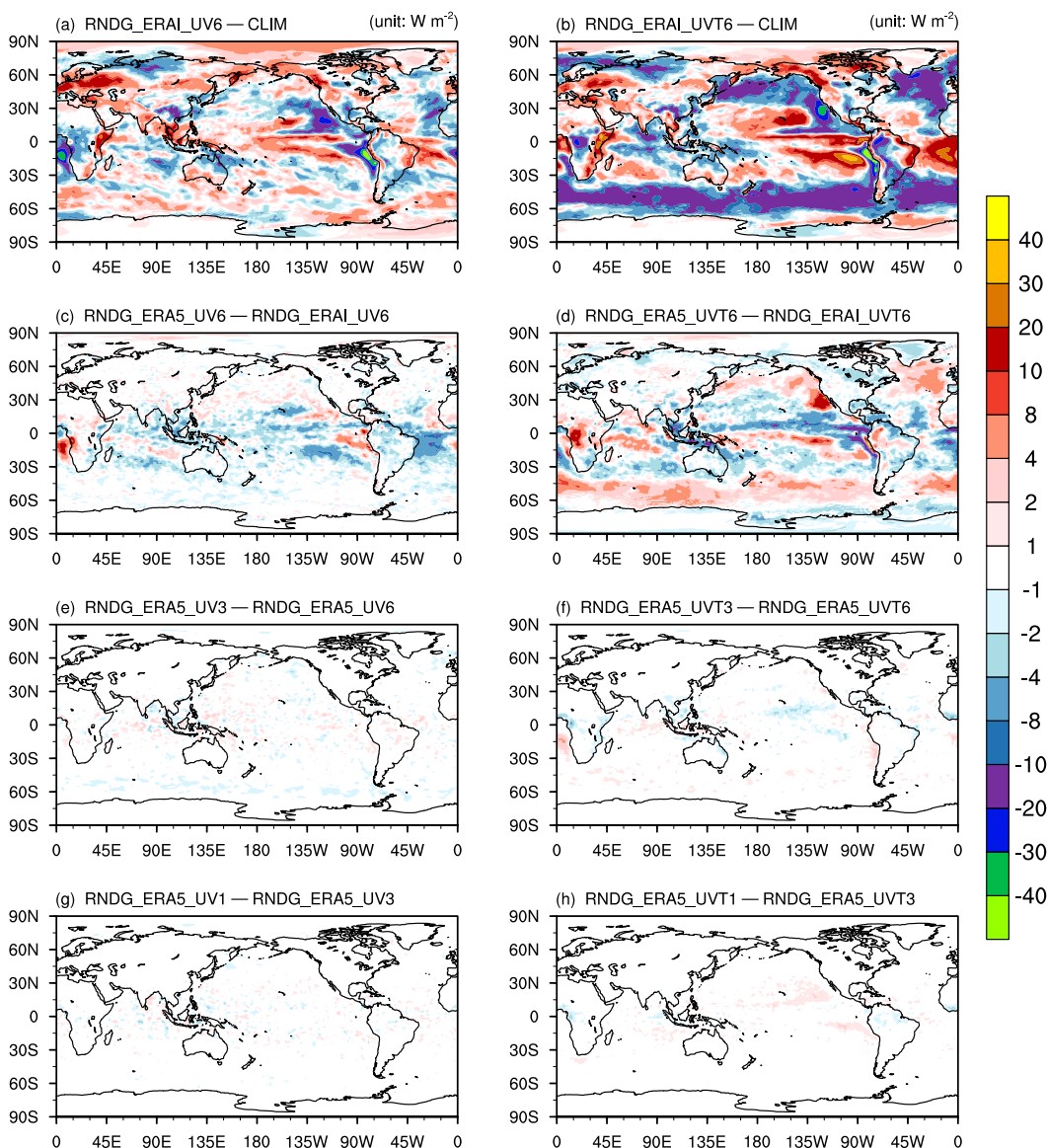

**Figure 6.** Differences in annual mean $CRE_{NET}$ (unit: $W\,m^{-2}$) in the PD simulations. The top row shows the differences between simulations nudged to ERA-Interim and the free-running baseline (CLIM). The second row shows differences between simulations nudged to ERA5 and ERA-Interim, both using 6-hourly constraining data temporally interpolated to every model time step. The third row shows the differences between simulations that interpolate 3-hourly versus 6-hourly reanalysis data to constrain the model. The last row is like the third row, but showing differences between two simulations nudged to hourly versus 3-hourly reanalysis data interpolated to model time steps. The left and right columns correspond to wind-only nudging and wind-and-temperature nudging, respectively. Details of the simulation setup can be found in Section 2.3 and Table 1.

smaller than the differences between either reanalysis and the free-running EAMv1 simulations. (As an example, the annual mean zonal mean pressure-latitude cross-sections of air temperature differences are shown in Fig. C3). Increasing the data frequency from 6-hourly to 3-hourly can lead to local changes of 1 to 4 W m $^{-2}$ in $CRE_{NET}$. These magnitudes are similar to what we have seen in Fig. 4a-d for the simulations nudged to CLIM. Further increasing the data frequency to hourly only introduces negligible changes, again similar to what we have seen in simulations nudged to CLIM (Fig. 4e-f).

A large number of model output variables have been examined in addition to CF, where we consistently see the differences between ERA-Interim-nudged and ERA5 nudged simulations being substantially smaller than the differences between nudged runs and CLIM, although the magnitudes are non-negligible in some regions. We also consistently see the fact that increasing data frequency from 6-hourly to 3-hourly can lead to discernible changes locally while a further increase to hourly data no longer leads to substantial differences. The impacts of data frequency on the simulated global averages are generally very small (see Table S3).

## 4.2 Global and regional weather events

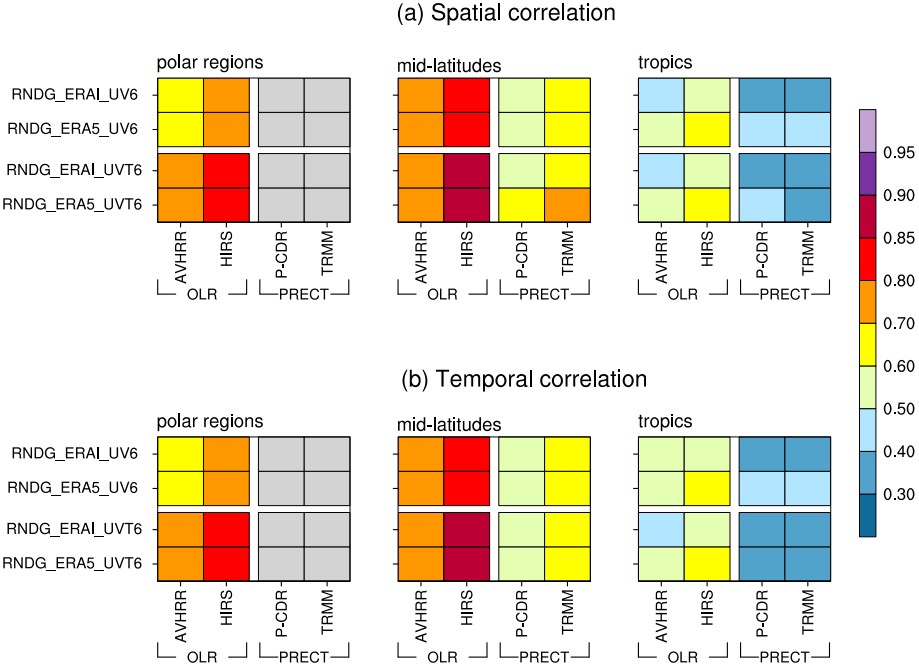

**Figure 7.** Anomaly correlations between the simulated and observed OLR and precipitation: (a) annual mean spatial correlations; (b) spatially averaged temporal correlations. Different latitude bands are examined separately: the Polar Regions ($60-90°$S, $60-90°$N), the mid-latitudes ($30-60°$S, $30-60°$N), and the tropics ($20°$S$-20°$N). The physical quantities and sources of observational data are indicated along the x-axis in each panel. All correlations were calculated using anomalies with respect to monthly averages. Gray boxes indicate missing values resulting from observational data being unavailable. The simulation setups are described in Section 2.3 and Table 1.

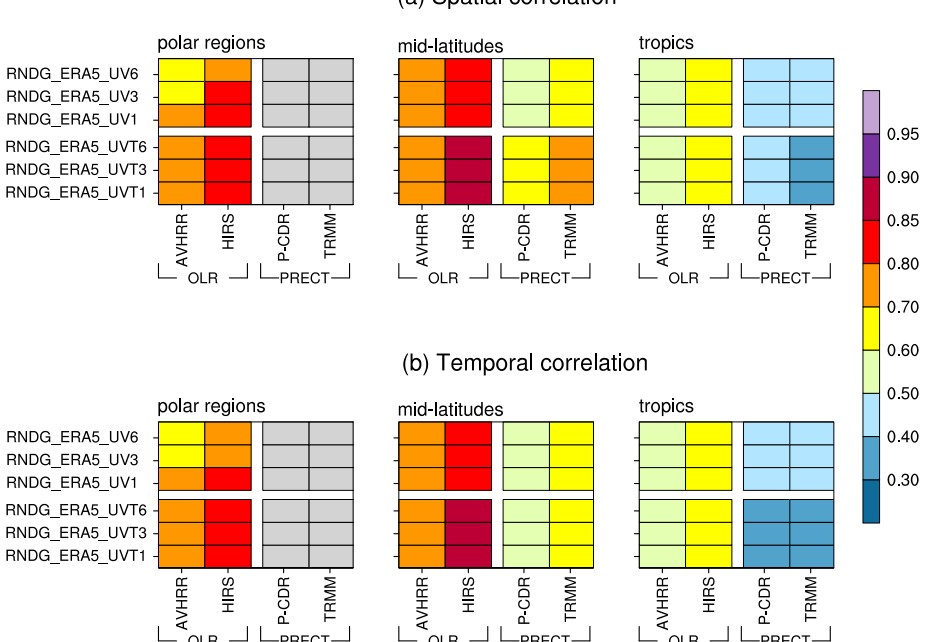

**Figure 8.** Similar to Fig. 7 but for simulations using different nudging-data frequencies (6-hourly, 3-hourly, or hourly). All simulations shown in this figure were nudged to the ERA5 reanalysis. Simulations shown in the top three rows of (a) and (b) used the wind-only nudging, while simulations shown in the bottom three rows of (a) and (b) used the wind-and-temperature nudging.

To evaluate the simulation of large-scale weather events, we follow the procedure used for Fig. 5 in Sun et al. (2019) and examine the anomaly correlation between nudged simulations and the observations. Here, an anomaly is defined as the deviation of a simulated or observed quantity from the corresponding (simulated or observed) monthly average at the same geographical location. We first examined the anomaly correlation between the nudged simulations and the corresponding reanalysis (ERA-Interim or ERA5) for temperature, specific humidity, as well as horizontal and vertical winds at various pressure levels. The results were found to be very similar to those presented in Fig. 5 in Sun et al. (2019). ERA-Interim and ERA5 nudged simulations show similar correlations to the corresponding reanalyses (see Fig. S1).

Since the discussion in this section focuses on comparing the hindcast skill of the ERA-Interim-nudged and ERA5-nudged simulations, we present in Figs. 7 and 8 an evaluation against global and regional-scale satellite retrievals of outgoing long-wave radiation (OLR) and surface precipitation rate. Panel (a) in each figure shows the annual average of spatial correlations in different latitude bands; panel (b) in each figure shows the spatially averaged temporal correlations of the anomalies. In Fig. 7, the two upper rows in each panel compare ERA-Interim-nudged and ERA5-nudged simulations that used wind-only nudging, while the lower rows compare simulations that also used temperature nudging. Figure 8 compares ERA5-nudged simulations that used different data frequencies. The EAM-simulated OLR is compared with National Oceanic and Atmospheric Administration's (NOAA's) daily retrievals from the High Resolution Infrared Radiation Sounder (HIRS, Lee et al., 2007) and the

Advanced Very High Resolution Radiometer (AVHRR, Stowe et al., 2002). The simulated total precipitation rate is compared with 3-hourly data from the Tropical Rainfall Measuring Mission (TRMM) 3B42V7 product (Huffman et al., 2007; Huffman and Bolvin, 2013) and daily data from the Precipitation Estimation from Remotely Sensed Information using Artificial Neural Networks-Climate Data Record (labeled as "P-CDR" in figures here, Ashouri et al., 2015). Further details of the datasets and the comparison procedure can be found in Sun et al. (2019).

The anomaly correlations shown in Fig. 7 indicate that the correlations in the high- and mid-latitude regions are very similar between the ERA5 and ERA-Interim nudged simulations regardless of whether temperature is constrained. In the low latitudes (20 °S to 20 °N), the correlations are higher when ERA5 is used as the constraining data, in terms of both OLR and precipitation, and both with or without temperature nudging. Figure 8 indicates that the changes associated with higher data frequency are small for the annual or regional averages shown here.

Figure 9 evaluates the simulated zonal and temporal propagation of meridionally averaged precipitation rate in boreal spring (March to May) of 2010 over the tropical Pacific Ocean (10$^o$S–10$^o$N, 60$^o$E–90$^o$W, upper row) and North America (25$^o$N–50$^o$N, 150$^o$E-60$^o$W, lower row). Panels (a) and (d) are Hovmöller diagrams plotted from the TRMM data. The bar charts show the correlation between the Hovmöller diagram of TRMM data and the corresponding Hovmöller diagrams plotted from various nudged simulations. Consistent with the anomaly correlations shown in Figs. 7 and 8, in the tropics we see a clear improvement in the simulated propagation of precipitation when ERA5 is used as the constraining data (Fig. 9b) while in the mid-latitudes there are no substantial differences between ERA-Interim-nudged and ERA5-nudged results (Fig. 9e). The impact of frequency of the constraining data is negligible (Fig. 9c, f). The same conclusions can be drawn if we use the root-mean-square error (RMSE) as the evaluation metric (cf. Figure C4), and if we change the evaluation to a different season (see Figs. S2 and S3).

As an aside, we note that the better precipitation hindcast skills in the mid-latitudes than in the tropics (Fig. 9c versus d) are consistent with the findings in Sun et al. (2019). The impact of constraining temperature appears to be negligible for the 2010 results shown here (Fig. 9c-d, solid fill versus hatching), while Sun et al. (2019) showed better precipitation hindcast skill with additional temperature nudging for spring 2011, especially in the tropics (see Figures 6 and 7 therein). This suggests that the role of temperature nudging can be case-dependent. Future evaluation in this aspect will be useful.

### 4.3 Comparison with the ARM data

To further assess the hindcast skill of the nudged simulations, we use the radiosonde observations collected by the US Department of Energy's Atmospheric Radiation Measurement (ARM) user facility. Radiosonde data are often considered to be reliable high-accuracy measurements (Milrad, 2017) and therefore can provide an objective evaluation of model simulations. Data from three ARM atmospheric observatories are selected to cover different climate regimes, including the Southern Great Plains (SGP) site over the mid-latitude land (https://www.arm.gov/capabilities/observatories/sgp), the North Slope of Alaska (NSA) site in the Northern Hemisphere polar region (https://www.arm.gov/capabilities/observatories/nsa), and the three central facilities at the Tropical Western Pacific site (Manus, TWPC1; Nauru, TWPC2; Darwin, TWPC3; https://www.arm.gov/capabilities/observatories/twp). To our knowledge, radiosonde measurements from these sites were not

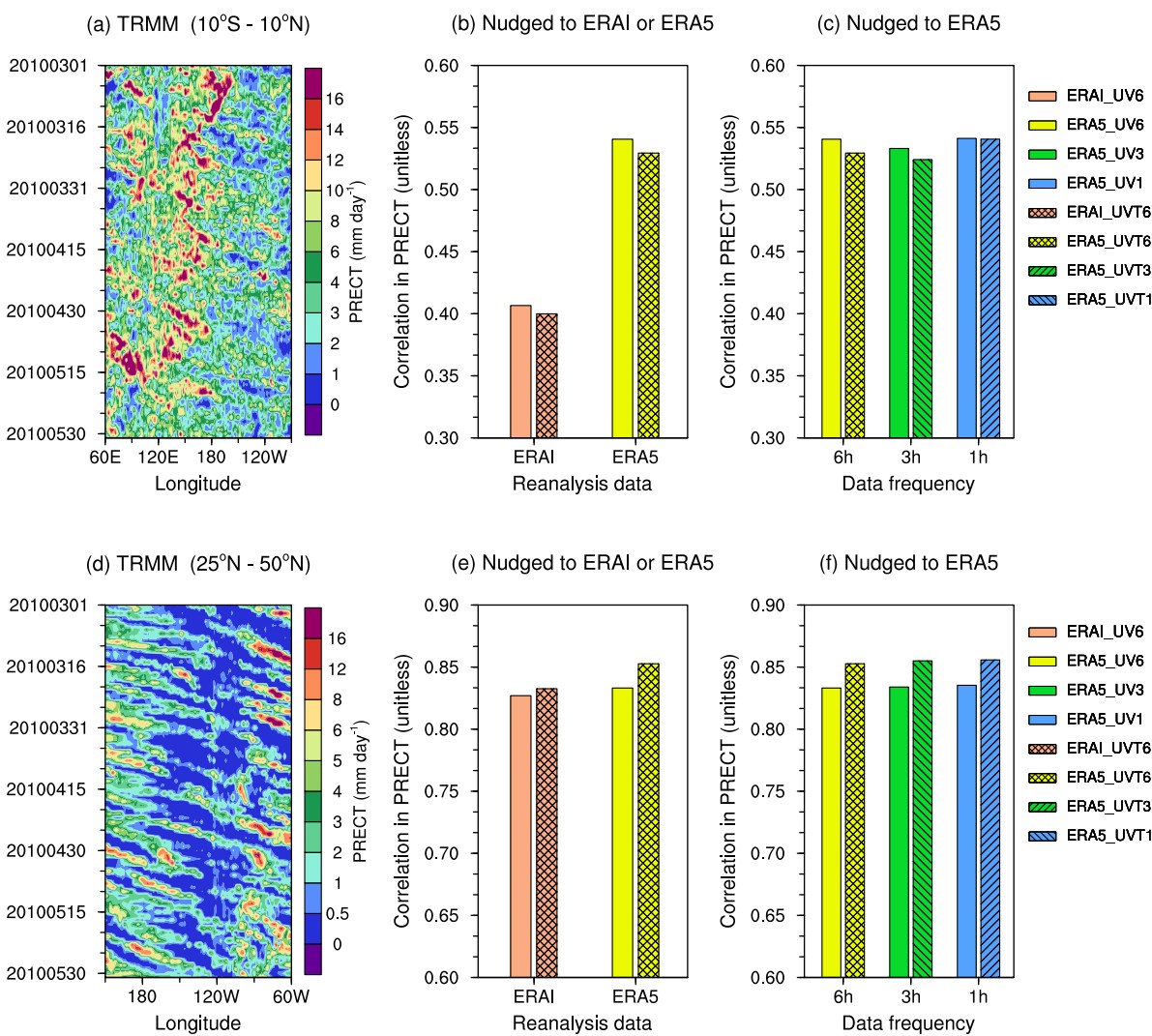

**Figure 9.** Evaluation of the spatio-temporal distribution of daily precipitation from 1 March to 31 May 2010 over the tropical Pacific Ocean ($10^o$S–$10^o$N, $60^o$E–$90^o$W, upper row) and North America ($25^o$N–$50^o$N, $150^o$E-$60^o$W, lower row). (a) and (d): Hovmöller diagrams of the meridionally averaged total precipitation rates (PRECT, unit: mm day$^{-1}$) from TRMM. The dates are labeled along the y axis. (b–c) and (e–f): correlations between a Hovmöller diagram derived from TRMM and the Hovmöller diagram derived from various nudged simulations. Panels (b) and (e) compare simulations using ERA-Interim or ERA5 as constraining data and with or without temperature nudging. Panels (c) and (f) compare simulations with U, V or U, V, and T nudged towards ERA5 but using 6-hourly, 3-hourly, and hourly reanalysis for the constraining data. All nudged simulations shown here used the sequence of calculations in Fig. 1b, so the prefix "RNDG_" is dropped to keep the legends short. The simulation setups are described in Section 2.3 and Table 1.

used in the data assimilation system producing the ERA reanalysis products, and hence can be considered to be independent data for the evaluation of the simulations nudged to ERA-Interim or ERA5.

The simulated temperature, relative humidity, and horizontal winds in January 2010 are evaluated against measurements collected in the same time period at SGP (Fig. 10, upper row), NSA (Fig. 10, lower row), and TWP (Fig. 11). The ERA-Interim reanalysis (black dashed lines in the figures) and ERA5 (black solid lines) are also included for comparison. The 6-hourly model output and reanalysis products were horizontally remapped to the locations of ARM sites using bilinear interpolation. For each of the meteorological quantities shown here, the root-mean-square errors (RMSEs) between the ERA-nudged simulations (or ERA analyses) and the ARM measurements were calculated with all available vertical profiles at the sites in January 2010. For each of the variables shown in Figs. 10 and 11 (i.e., T, RH, U, or V), the numbers of available vertical profiles at SGP, NSA, TWPC1, TWPC2 and TWPC3 were 121, 63, 49, 57 and 127, respectively. (ARM sites provide data four times a day at SGP and TWPC3, and twice a day at NSA, TWPC1 and TWPC2). The temporal correlations between EAM simulations or ERA analyses and the ARM measurements are shown in Fig. C5 and Fig. C6 in the Appendix.

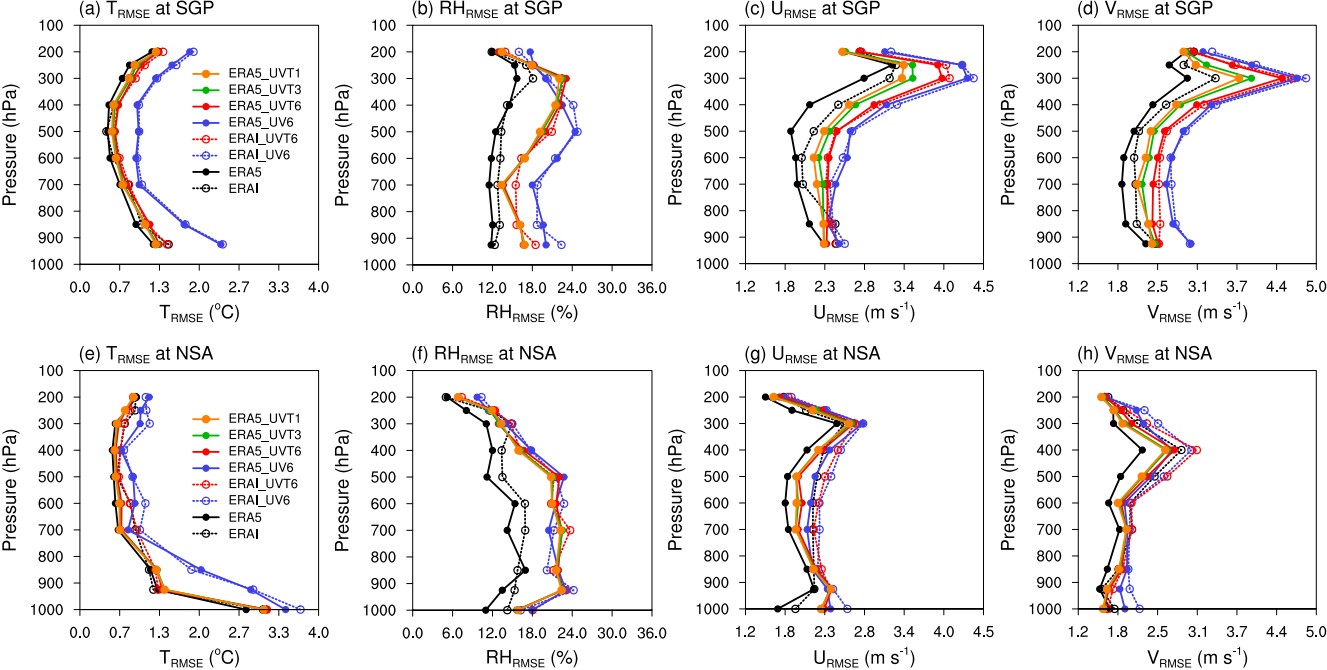

**Figure 10.** Comparison of two reanalysis products (ERA-Interim and ERA5, black lines) and various nudged simulations (colored lines) with ARM radiosonde measurements of January 2010 at SGP (upper row) and NSA (lower row). The four columns from left to right show the root-mean-square errors (RMSEs) in temperature (T, unit: °C), relative humidity (RH, unit: percent), zonal wind (U, unit: m s$^{-1}$) and meridional wind (V, unit: m s$^{-1}$), respectively. All nudged simulations shown here used the sequence of calculations in Fig. 1b, so the prefix "RNDG_" is dropped in this figure to keep the labels short. The simulation setups can be found in Section 2.3 and Table 1.

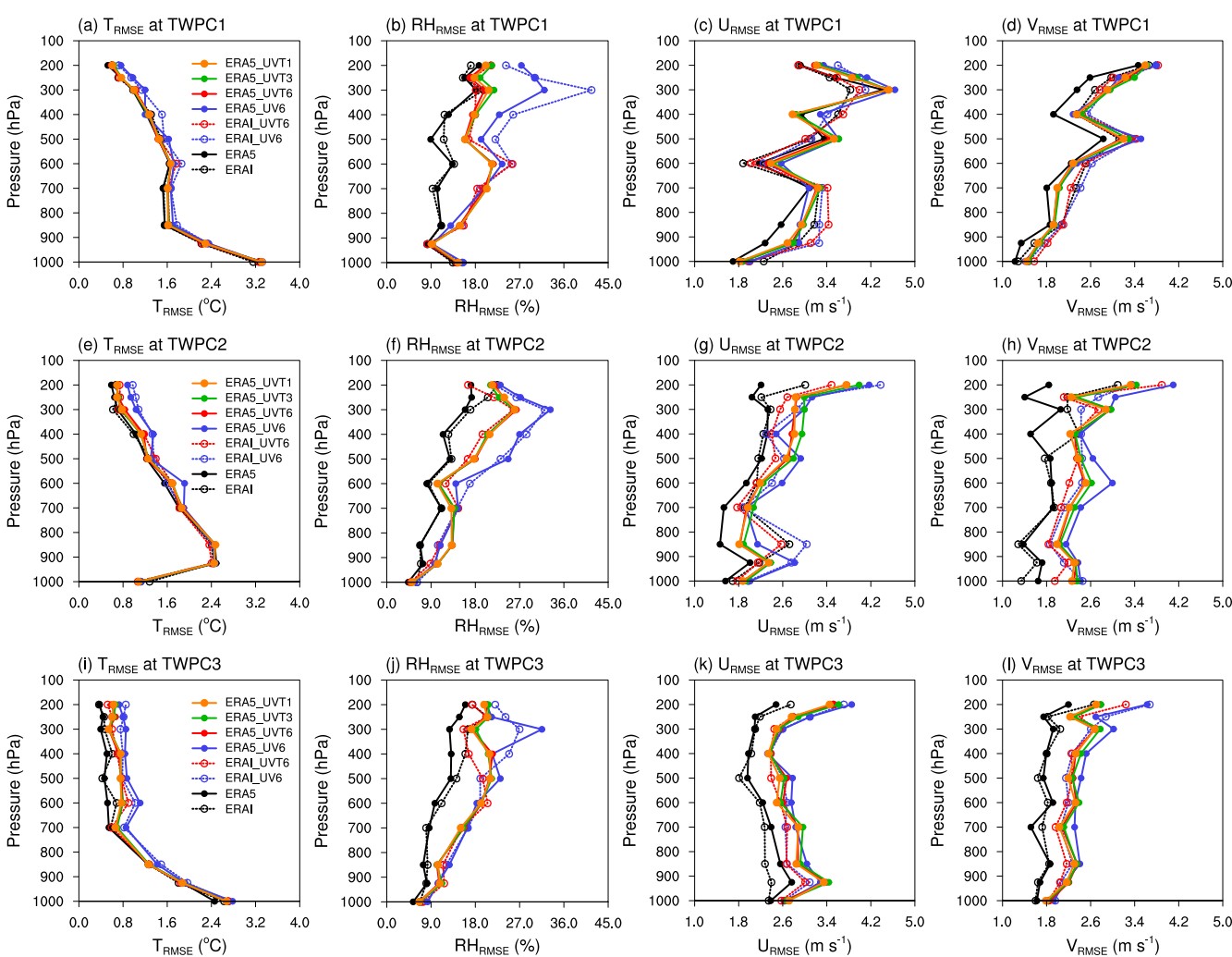

**Figure 11.** As in Fig. 10 but showing the RMSEs at the three central facilities of ARM's site, TWPC1–C3.

As expected, reanalyses (black lines in Fig. 10, Fig. 11, Fig. C5 and Fig. C6 ) show better agreement with the ARM radiosonde data compared to the nudged EAM simulations (colored lines). ERA5 (solid black in Fig. 10 and Fig. C5) is in general better than ERA-Interim (dashed black) at the mid-latitude SGP site and the high-latitude NSA site. For the three ARM TWP sites in the tropics, ERA5 is not always better than ERA-Interim. For example, ERA5's zonal wind field (U, solid black in Fig. 10d) shows larger RMSEs below 500 hPa compared to the ERA-Interim (dashed black).

The ERA-nudged simulations show good agreement with the ARM radiosonde measurements at the mid-latitude (SGP) and high-latitude (NSA) sites (Fig. 10). Compared to the ERA-Interim-nudged simulations (colored dashed lines), the ERA5-nudged simulations (colored solid lines) have slightly better hindcast skills. In addition, EAM simulations with temperature nudging (red, orange and green lines) show overall better hindcast skills, regardless of which ERA product was used as the constraining data. Slightly better hindcast skills for horizontal winds can be obtained by using the 3-hourly ERA5 data (green lines) for nudging instead of using 6-hourly data (red lines). Using 3-hourly (green lines) or hourly (orange lines) constraining data gives very similar results.

Consistent with the experiences reported in the literature (e.g., Jeuken et al., 1996; Sun et al., 2019), weather events in the tropics are less well constrained by nudging. Compared to the ARM SGP and NSA sites (Fig. 10, Fig. C5), the magnitude of the RMSEs in ERA-nudged simulation at three ARM TWP sites are in similar ranges (Fig. 11), while the temporal correlations are smaller at the tropical sites, especially for temperature and relative humidity (Fig. C6). At the tropical sites, we do not see systematic improvements when switching from EAM-Interim to EAM5 for the constraining data or when increasing the data frequency, although the UVT nudging (red, orange and green lines in Fig. 11 and Fig. C6) still provides better hindcast skills than the UV nudging (blue lines).

## 5    Estimation of the anthropogenic aerosol effects

Nudging has been recognized as a useful and computationally efficient technique for estimating the anthropogenic aerosol effects in global climate models (Kooperman et al., 2012; Zhang et al., 2014, 2016; Ghan et al., 2016; Liu et al., 2018). In this section, we evaluate the impact of nudging configuration on the estimated anthropogenic aerosol effects in EAMv1. It is of practical value to identify nudged simulations that are capable of providing estimates consistent with the results from free-running simulations, as the EAM developers have identified the anthropogenic aerosol effects to be one of the key aspects that need more attention in the future development and evaluation of the model (Golaz et al., 2019; Zhang et al., 2022). Similar to previous studies, we calculate the anthropogenic aerosol effects by contrasting a pair of EAMv1 simulations conducted with PD (year 2010) and PI (year 1850) emissions of the anthropogenic aerosols and precursors following the CMIP6 protocol (Eyring et al., 2016; Hoesly et al., 2018; Feng et al., 2020).

### 5.1    Results from the free-running EAMv1

As explained in Sect. 2.3 and summarized in group 1 of Table 1, we carried out 5 pairs of 1-year simulations without nudging (the "CLIM" runs) or with very weak nudging (simulations "CLIMp1"–"CLIMp4"). The 5-member mean, one-year mean,

globally averaged PD-PI difference in the TOA net radiative flux, $\Delta F_{NET}$, is about -1.7 W m$^{-2}$. The shortwave component is $\Delta F_{SW}$ = -2.4 W m$^{-2}$, and the longwave component is $\Delta F_{LW}$ = 0.7 W m$^{-2}$ (see Table S4). These numbers are consistent with the effective aerosol forcing estimates reported in Sect. 6.1 of Golaz et al. (2019). The PD-PI differences in the shortwave and longwave CRE in our CLIM ensemble are $\Delta CRE_{SW}$ = -1.7 W m$^{-2}$ and $\Delta CRE_{LW}$ = 0.6 W m$^{-2}$, respectively (Table S4), which gives a net effect of $\Delta CRE_{NET}$ = -1.1 W m$^{-2}$.

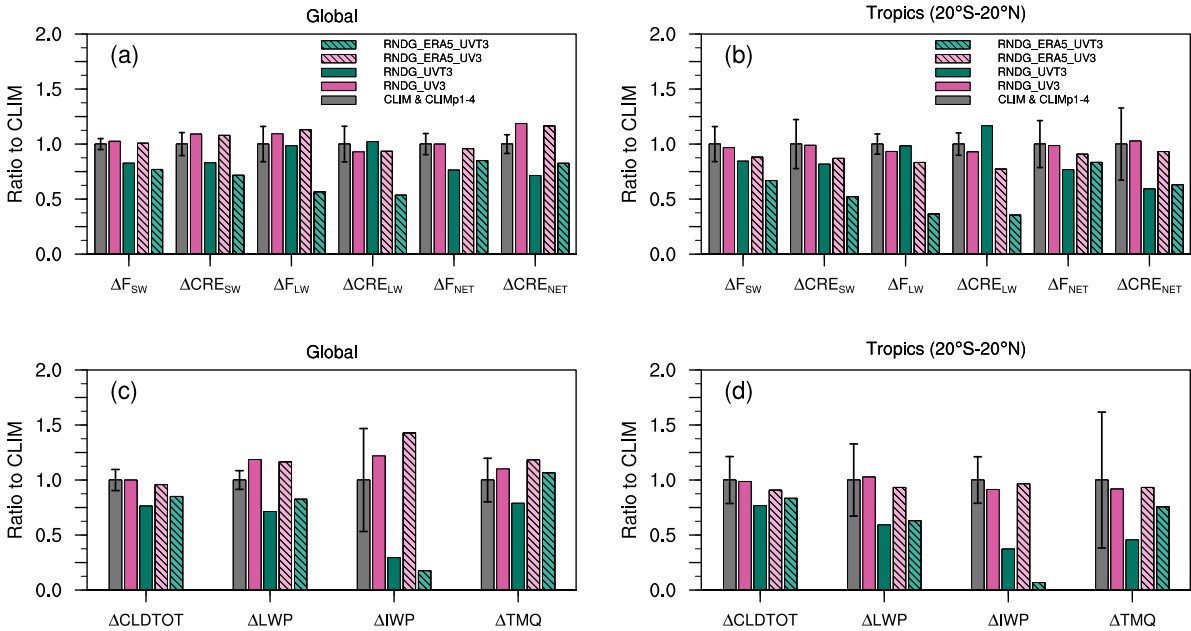

**Figure 12.** First row: global mean (a) and tropical mean (b) annually-averaged anthropogenic aerosol effects on TOA radiative fluxes and CRE estimated by free-running and nudged EAM simulations. The variable names noted along the x-axis are explained in Sect. 2.3. Simulation short names shown as legends are explained in Table 1 and Sect. 2.3. Second row: as in the first row but for the PD-PI difference in total cloud fraction ($\Delta$CLDTOT), liquid water path ($\Delta$LWP), ice water path ($\Delta$IWP) and total precipitable water ($\Delta$TMQ). All values have been normalized by the ensemble mean of CLIM and CLIMp1-4. The thick whiskers attached to the gray bars indicate the two-standard-deviation ranges of the 5-member CLIM ensemble. The non-normalized data can be found in Tables S4 and S5 in the supplemental materials.

## 5.2   Impacts of temperature nudging

In Fig. 12, we compare various configurations of the nudged simulations against the CLIM ensemble in terms of the annual mean PD-PI differences averaged over the globe (left column) or the tropics (20°S–20°N, right column). All results are normalized by the ensemble mean of CLIM. The thick black whiskers attached to the gray bars indicate the two-standard-deviation ranges of the CLIM ensemble. The non-normalized data can be found in Tables S4 and S5 in the supplemental materials. All

of the nudged simulations shown in the figure used the revised sequence of calculations and 3-hourly constraining data. Two of the nudged simulations were constrained by ERA5 (the colored bars with hatch filling) and the other two by CLIM PD

(the colored bars with solid filling). Our first focus is to compare simulations conducted using UV-nudging with those using UVT-nudging.

Generally speaking, we can expect the nudging of temperature to have three potential impacts.

– First, if a model has significant and systematic temperature biases relative to reanalysis, then nudging temperature towards reanalysis will introduce a mean bias correction, which can trigger responses of the model atmosphere in the nudged simulations and consequently cause differences (compared to the free-running simulations) in the aerosol effects.

– Second, it is worth noting that the anthropogenic aerosol effects we are trying to estimate here is the ERF, which, by
definition, is the changes in the TOA radiative fluxes when all physical variables in a climate model are allowed to respond to perturbations except for those concerning the ocean and sea ice (Myhre et al., 2014). Because anthropogenic aerosols have significant cooling effects, when a pair of simulations with PD and PI emissions are both nudged to the same temperature fields, the aerosol-induced temperature differences between PD and PI conditions can be substantially reduced regardless of which constraining data is used. This can then leads to a significant suppression of the simulated
aerosol effects and hence inconsistency between the temperature-nudged and the free-running simulations.

– Third, when the constraining temperature data is provided at a relatively low frequency, we might encounter situations similar to those illustrated in Fig. 3, namely the linearly interpolated constraining data can fail to capture high-frequency variations in temperature and hence causing responses in the simulated atmospheric state. The impact of such aliasing is expected to be discernible only if the high-frequency modes of variability have substantial roles in determining the
anthropogenic aerosol effects.

We show in Sect. 5.3 that the third effect (aliasing) is indeed small in EAMv1. In the remainder of this subsection, we focus on quantify the first two effects listed above.

Panel (a) of Fig. 12 shows the global mean PD-PI differences in the TOA fluxes and CRE, and panel (b) of the figure shows the averages over the tropics. The second row of Fig. 12 shows the PD-PI differences in the global and tropical mean total cloud
fraction ($\Delta$CLDTOT), cloud liquid and ice water path ($\Delta$LWP and $\Delta$IWP), and total precipitable water ($\Delta$TMQ). All results shown in this figure have been normalized by the ensemble mean of CLIM. The non-normalized data can be found in Table S4 and S5. Keeping in mind the ensemble spread of the CLIM simulations, we see that the estimates obtained with UV-nudging (pink bars) are consistent with the estimates from CLIM, while the estimates obtained with UVT-nudging (green bars) show statistically significant deviations from CLIM. This is true for both the ERA5-nudged and CLIM-nudged simulations.

### 5.2.1 Mean bias correction

The impact of mean bias correction caused by nudging EAM's temperature to reanalysis can be evaluated by comparing simulations RNDG_ERA5_UVT (hatched teal bars in Fig. 12) and RNDG_UVT (solid teal bars) against CLIM (gray bars and whiskers). Overall, the hatched teal bars show larger deviations than the solid teal bars, suggesting that the mean bias correction has significant impacts.

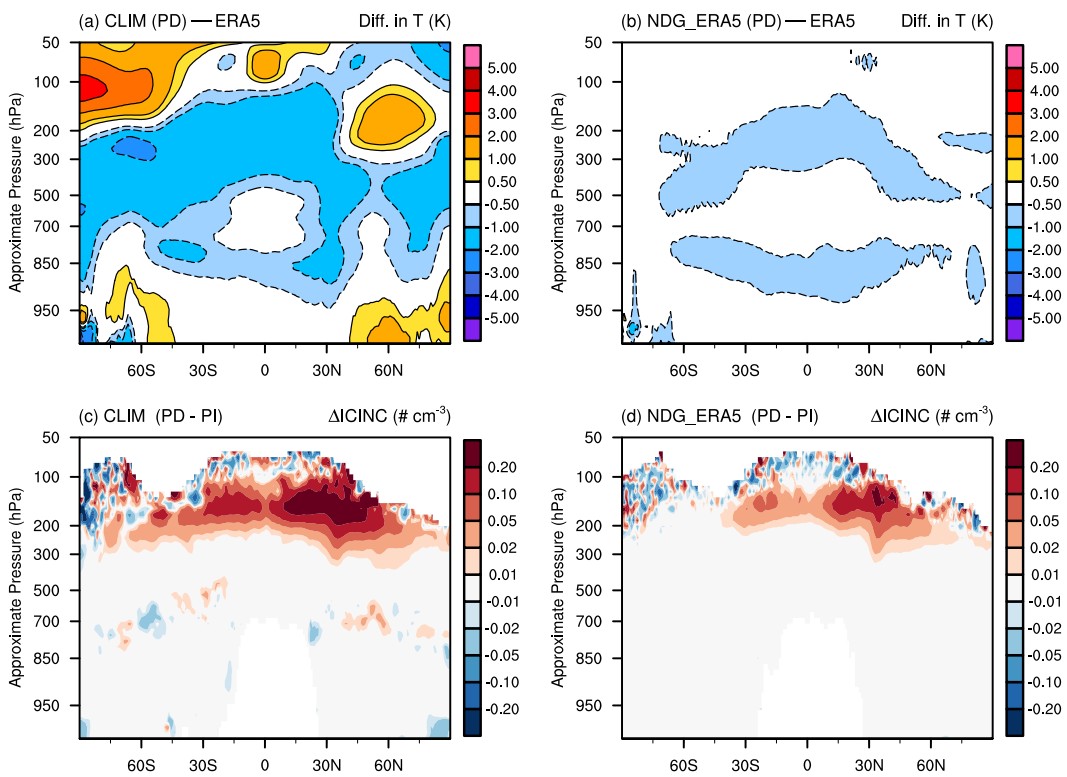

**Figure 13.** Upper row: zonal and annual mean differences in temperature (T, unit: K) between the CLIM PD simulation and the ERA5 reanalysis (panel a), and between a nudged PD simulation and ERA5 (panel b). The nudged simulation is labeled as "NDG_ERA5 (PD)" for brevity in panel b; it correspond to the simulation RNDG_ERA5_UVT3 in Table 1 performed with PD emissions of aerosols and precursors. Lower row: PD-PI differences of the in-cloud ice number concentration ($\Delta$ICINC, unit: # cm$^{-3}$) derived from the free-running simulations (CLIM, panel c) and from EAMv1 simulations with UVT-nudging towards ERA5 (i.e. RNDG_ERA5_UVT3, panel d). Details of the simulation setups can be found in Sect. 2.3 and Table 1.

Among the quantities shown in Fig. 12, the tropical mean $\Delta F_{LW}$, $\Delta CRE_{LW}$, $\Delta IWP$ and $\Delta CLDTOT$ show the largest reduction when temperature is nudged to ERA5 instead of CLIM (panels b and d). This can be explained by the zonal and annual mean temperature differences ($\Delta T$) and in-cloud ice number concentration differences ($\Delta$ICINC) shown in Fig. 13. Fig. 13a suggests that EAMv1's climatology, when compared to ERA5, features cold biases on the order of 1-2 K in the upper troposphere over the tropical and mid-latitude regions where small ice crystals are often formed through homogeneous ice nu-

cleation. These small ice crystals are known to have a large impact on the simulated CRE. Nudging EAM's temperature towards ERA5 leads to a warmer base state and weakened homogeneous ice nucleation compared to CLIM (Fig. C7b). Consequently, the PD-PI changes in aerosol and precursor emissions cause substantially smaller $\Delta$ICNIC compared to CLIM (Fig. 13d versus c), which explains the significant reduction in $\Delta F_{LW}$ and $\Delta CRE_{LW}$ shown as hatched green bars in Fig. 12a–b. This reasoning is consistent with the finding in Zhang et al. (2014) that temperature nudging in EAMv1's predecessor model CAM5 led to a

substantial decrease in the ice cloud amount and a weaker impact of anthropogenic aerosols on longwave radiation. Although nudging temperature towards reanalysis can improve the agreement between the observed and EAM-simulated long-term climate, it significantly changes the simulated cloud amounts and properties as well as the CRE in EAMv1, and hence does not achieve the goal of obtaining estimates of the anthropogenic aerosol effects that are consistent with those in the free-running EAM simulations.

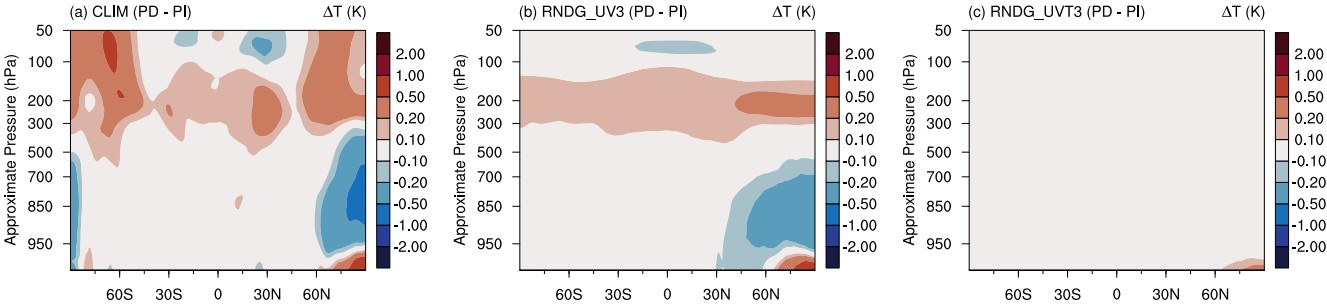

**Figure 14.** PD-PI differences in temperature (ΔT, unit: K) in the free-running simulations (CLIM, panel a), the EAM simulations nudged to U and V from CLIM PD (panel b), and the EAM simulations nudged to U, V, and T from CLIM PD (panel c). Further details of the simulation setup can be found in Sect. 2.3 and Table 1.

### 5.2.2 Suppression of temperature changes

The impact of nudging PD and PI simulations with the same temperature data and hence suppressing the ERF can be evaluated by contrasting the two simulations RNDG_UV3 and RNDG_UVT3 (solid bars in pink and teal, in Fig. 12). For reference, we shown in Fig. 14 the zonal mean PD-PI temperature difference in CLIM and in the simulations nudged to CLIM PD. In the free-running simulations, the increased emissions of anthropogenic aerosols and precursors lead to considerable cooling in the

460 lower troposphere in the Northern Hemisphere mid- and high-latitude regions as well as warming in the upper troposphere (Fig. 14a). These temperature changes are captured by the simulations nudged to U and V from CLIM PD (Fig. 14b), while the simulations with temperature nudging show negligible ΔT (as expected).

Consistent with the ΔT shown in Fig. 14, we see that temperature nudging has considerable impacts on the simulated PD-PI differences in LWP, IWP, CLDTOT (Fig. 12 lower row, teal versus pink solid bars). The impact on global mean ERF is more

clearly seen in the shortwave component (Fig. 12a), possibly due to the cloud changes caused by the lack of cooling (compared to CLIM) in the mid- and high-latitude lower troposphere when temperature is nudged (Fig. 14c versus b).

### 5.2.3 Combined effects

The NDG_ERA5_UVT3 configuration is affected by both the mean state correction and the suppression of PD-PI temperature changes, hence it is not surprising that the results (hatched teal bars in Fig. 12) deviate most substantially from

470 CLIM compared to all other nudged simulations shown in the figure. For global averages (Fig. 12a), the $\Delta F_{SW}$ estimated by

NDG_ERA5_UVT3 is about 25% lower than CLIM, and the $\Delta F_{LW}$ is about 50% lower than CLIM (see also Table S4). In the topics (Fig. 12b), there is a 33% decrease in the estimated $\Delta F_{SW}$ and a 63% decrease in $\Delta F_{LW}$ (see also Table S5). Large discrepancies between NDG_ERA5_UVT3 and CLIM are also seen in $\Delta CRE_{SW}$ and $\Delta CRE_{LW}$ (Fig. 12 upper row) and in the cloud-related fields (Fig. 12 lower row).

It is worth noting that the discrepancies in $\Delta F_{NET}$ might appear to be not as large. For example, we see a 15% difference in the global mean in (Fig. 12a and Table S4) and a 17% difference in the tropical average (Fig. 12b and Table S5). But the smaller differences in the net fluxes are results of the cancellation of large changes in the shortwave and longwave components caused by temperature nudging.

     Overall, our results discussed above suggest that nudging the horizontal winds but not temperature is the preferred simulation
configuration for estimating the anthropogenic aerosol effects, which is consistent with the results reported in Zhang et al. (2014). When the PD-PI simulations are constrained by the same temperature data, nudging can suppress the adjustments to aerosol forcing; When the constraining data comes from a different model (e.g., reanalysis), one can get an additional effect, namely the mean bias corrections can significantly modify the simulated clouds and their responses to aerosol forcing. In both cases, temperature nudging can lead to estimates of the anthropogeneic aerosol effects that are significantly different from the
results obtained from the free-running simulations, which is undesirable for many model development and evaluation studies.

### 5.3 Impacts of the frequency of constraining data

In Fig. 15, we evaluate the impacts of the frequency of the constraining data used in nudged simulations. At least for the annual mean effects, the results obtained from simulations using 6-hourly constraining data (orange bars in the figure) are very similar to those obtained using 3-hourly constraining data (blue bars), regardless of whether UV-nudging (Fig. 15, upper row)
or UVT-nudging (Fig. 15, lower row) is used. The small impact of constraining data frequency on global and tropical mean $F_{aer}$ estimates is expected. As shown in Section 3.2, the impact of constraining data frequency on present-day simulations is sizable only in limited regions where strong diurnal variations exist. Therefore, using 6-hourly constraining data in nudged simulations is sufficient for estimating the global and annual mean $F_{aer}$.

### 6 Conclusions

Nudging has been widely used in the development and evaluation of global and regional atmospheric models. In this work, we further improved the nudging implementation in EAMv1 compared to the work of Sun et al. (2019) and evaluated the impact on the climate representativeness, the hindcast skill of nudged simulations, and the estimation of anthropogenic aerosol effects.

     The study was motivated by an unresolved issue in Sun et al. (2019), namely a nudged EAMv1 simulation constrained by EAMv1's own meteorology showed non-negligible local deviations from the baseline, with annually-averaged $CRE_{SW}$
(SWCF) discrepancies as large as 4–8 W m$^{-2}$ over some of the subtropical marine stratocumulus and trade cumulus regions. Two reasons were identified: First, EAMv1 writes out meteorological fields (from a baseline simulation) for nudging before the radiation parameterization, but the nudging tendency is calculated at a different location in the time integration loop, i.e., after

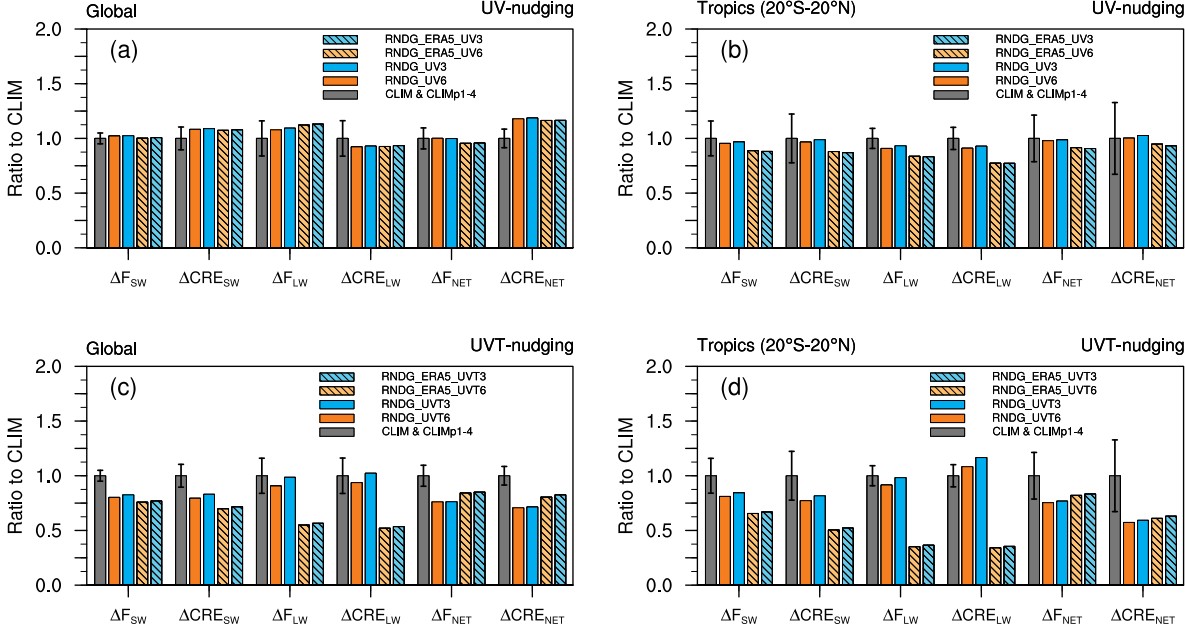

**Figure 15.** Global mean (left column) and tropical mean (right column) annually-averaged anthropogenic aerosol effects on TOA radiative fluxes and CRE estimated by free-running EAM simulations (gray bars) and nudged runs (colored bars). The variable names noted along the x-axes are explained in Sect. 2.3. Simulation short names shown as legends are explained in Table 1 and Sect. 2.3. All values shown here have been normalized by the ensemble mean of the free-running simulations (i.e., CLIM and CLIMp1-4). The thick whiskers attached to the gray bars indicate the two-standard-deviation ranges of the 5-member free-running ensemble. Nudged simulations in the upper row (panels a and b) used UV-nudging; those in the lower row (panels c and d) used UVT-nudging. Orange and blue correspond to 6-hourly and 3-hourly constraining data, respectively. Hatched bars correspond to simulations nudged to ERA5, and the bars with solid color fill correspond to simulations nudged to the CLIM PD simulation.

the dynamical core. This inconsistency introduced an unintended contribution to the nudging tendency that was proportional to the effects of deep convection, shallow convection, and cloud microphysics on the simulated atmosphere (Section 3.1).

Second, the EAM-simulated winds and temperature in the lower troposphere were found to have high-frequency modes with non-negligible magnitudes. For example, the zonal wind in the Peruvian stratocumulus region was found to have a prominent 12-hour cycle. Such variations cannot be properly captured by a 6-hourly sampling frequency, hence resulting in significant aliasing issues with the constraining data used for nudging (Section 3.2). We showed that by moving the calculation of nudging tendency to the same location as data output (Fig. 1b) and by increasing the frequency of constraining data to 3-hourly, one

could largely remove the discrepancies between a 1° free-running EAMv1 simulation and a 1° nudged simulation constrained by EAM's own meteorology. Further increasing the data frequency to hourly only provided marginal improvements. For future studies that nudge EAM towards its own meteorology, we recommend using the revised implementation and the 3-hourly constraining data for 1° simulations. Whether simulations performed at higher horizontal resolutions can benefit from higher

data frequency remains to be investigated. For users of EAM, we have provided in Table A1 the nudging-related namelist settings for two of the simulations discussed in this paper to demonstrate how to turn on the revised sequence of calculations and change the constraining data frequency.

The abovementioned improvements further motivated us to investigate the potential benefits of using the ERA5 reanalysis, which is available at a higher frequency compared to ERA-Interim, for nudged hindcast simulations. In terms of the annual mean fields, there were discernible but small regional changes when switching from ERA-Interim to ERA5 or changing the constraining data frequency when using ERA5. The impacts on global mean climate were found to be small (Section 4.1). Satellite retrievals of OLR and precipitation were used to evaluate the model's skill in capturing real weather events. When ERA5 was used instead of ERA-Interim, the simulated OLR and precipitation were significantly improved, especially in the tropics. We also evaluated the nudged simulations using radiosonde measurements from several ARM sites in different climate regimes. At the SGP and NSA sites, the simulated horizontal winds, temperature, and relative humidity were systematically improved when replacing ERA-Interim with ERA5 and when using higher-frequency nudging data. Significant improvements were also seen in the mid- and high-latitude ARM sites. At the tropical sites (TWPC1, TWPC2, and TWPC3), the improvements were not as significant. At SGP and NSA, nudging winds and temperature together was found to further improve the hindcast skill of the simulations. Overall, the good agreement in the simulated and observed meteorological conditions provides a good basis for possible future studies that use ARM measurements to help identify parameterization deficiencies and improve the representation of cloud and aerosol-related atmospheric processes in EAM.

Last but not least, we evaluated the impact of nudging on the estimated anthropogenic aerosol effects. Results show that the frequency of the constraining data has negligible impacts on the estimated global and tropical averages of annual mean TOA fluxes and CRE, while the impact of temperature nudging is large. Similar to conclusions from earlier studies, we recommend nudging the horizontal winds but not the air temperature when attempting to obtain estimates of the aerosol effects that are consistent with the estimates from free-running simulations. The reason is twofold: when the simulations forced by PD and PI emissions are constrained by the same meteorological fields, nudging temperature can strongly suppress the temperature responses to the aerosol perturbations and consequently affect other atmospheric adjustments; when the constraining meteorology comes from a different model (e.g., reanalysis), one can get an additional effect, namely the effective mean bias corrections introduced by temperature nudging can significantly modify the simulated clouds and their responses to aerosol perturbations. In both cases, temperature nudging can lead to estimates of the anthropogenic aerosol effects that are significantly different from the results obtained from the free-running simulations, which is undesirable for many model development and evaluation studies. Our results from EAMv1 showed that when temperature in the PD and PI simulations were both nudged towards a free-running PD simulation, the shortwave component of the aerosol effects on TOA radiative flux and CRE was significantly underestimated. When the PD and PI simulations were nudged towards the PD temperature from the ERA5 reanalysis, both the shortwave and longwave components of the aerosol effects on TOA radiative flux and CRE were significantly underestimated. While the percentage discrepancies in the net TOA flux and CRE appeared to be considerably smaller, this was the result of the cancellation of large discrepancies in the shortwave and longwave components. In contrast, nudging horizontal winds but not

temperature provided estimates that were reasonably consistent with the results from the free-running simulations, regardless of whether the nudged simulations were constrained by ERA5 or EAM's own meteorology.

We note that the 1° configuration of EAMv1 was used in this study. Due to the relatively coarse grid spacing in our simulations, the benefits of the high temporal and spatial resolutions of the ERA5 data might not have been fully revealed. As pointed out by Jeuken et al. (1996), the linear temporal interpolation in nudging can become more questionable for higher-resolution simulations as more short time scale processes are resolved. Also, compared to ERA-Interim, ERA5 can provide more accurate meteorological variables at finer spatial scales, so the ERA5-nudged simulation might perform even better at high resolutions

than seen here in the 1° simulations. The high-resolution configuration of EAMv1 was substantially more expensive, and hence not used in this study. The newer version of EAM released in September 2021, EAMv2, has become significantly less expensive thanks to the use of a different physics grid (Hannah et al., 2021) and a more efficient large-scale advection scheme (Bradley et al., 2019). Hence, it will be useful to further explore nudged EAM simulations at higher resolutions.

# Appendix A: Namelist examples for nudged EAMv1 simulations

**Table A1.** Nudging-related namelist variables used in two simulations, DNDG_ERAI_UVT6 and RNDG_ERA5_UVT3. The simulation DNDG_ERAI_UVT6 used the sequence of calculations in the default EAMv1 (see Fig. 1a) and 6-hourly constraining data from ERA-Interim. RNDG_ERA5_UVT3 used the revised sequence of calculations shown in Fig. 1b and 3-hourly constraining data from ERA5. The namelist variables used for changing the sequence of calculations and constraining data frequency are highlighted in boldface. Further details of the simulation setup can be found in Sect. 2.3 and Table 1.

| Namelist variable | DNDG_ERAI_UVT6 | RNDG_ERA5_UVT3 |
| --- | --- | --- |
| nudge_model | .True. | .True. |
| nudge_method | 'Linear' | 'Linear' |
| nudge_currentstep | .False. | .False. |
| **Nudge_loc_physout** | **.False.** | **.True.** |
| nudge_tau | 6.0 | 6.0 |
| model_times_per_day | 48 | 48 |
| **nudge_times_per_day** | **4** | **8** |
| nudge_ucoef | 1.0 | 1.0 |
| nudge_uprof | 1 | 1 |
| nudge_vcoef | 1.0 | 1.0 |
| nudge_vprof | 1 | 1 |
| nudge_tcoef | 1.0 | 1.0 |
| nudge_tprof | 1 | 1 |
| nudge_qcoef | 0.0 | 0.0 |
| nudge_qprof | 0 | 0 |
| nudge_pscoef | 0.0 | 0.0 |
| nudge_psprof | 0 | 0 |
| nudge_path | './ERA-Interim/' | './ERA5/' |
| nudge_file_template | 'interim_se_%y-%m-%d-%s.nc' | 'era5_ne30L72_%y-%m-%d-%s.nc' |
| nudge_file_ntime | 1 | 1 |
| nudge_beg_year | 2009 | 2009 |
| nudge_beg_month | 10 | 10 |
| nudge_beg_day | 1 | 1 |
| nudge_end_year | 2011 | 2011 |
| nudge_end_month | 1 | 1 |
| nudge_end_day | 1 | 1 |

 **Appendix B: Notes for Fig. 5**

The physical quantities labeled with short names along the x-axis in Fig. 5 are explained below in Table B1. The error metrics shown in Fig. 5a–b are the relative differences in the simulated global mean annual averages. The error metrics shown in Fig. 5c–d are the relative differences in the annual mean global patterns (i.e., geographical distributions). Following Wan et al. (2021), a relative difference in the global mean annual average is normalized by the global mean annual average from CLIM. A relative difference in the annual mean global pattern is defined as the centered root-mean-square (RMS) difference between the pattern in the test simulation and the pattern in the reference simulation, normalized by the RMS of the pattern in the reference simulation. A "pattern" here represents the annal mean, global, geographical distribution of a physical quantity.

**Table B1.** List of EAM output variables used for the evaluation shown in Fig. 5.

| Physical quantity | EAM output |
|---|---|
| Surface longwave downwelling flux | FLDS |
| Surface net longwave flux | FLNS |
| TOA upward longwave flux | FLUT |
| TOA clearsky upward longwave flux | FLUTC |
| Surface net shortwave flux | FSNS |
| TOA net shortwave flux | FSNTOA |
| TOA clearsky net shortwave flux | FSNTOAC |
| Longwave cloud radiative effect | LWCF |
| Shortwave cloud radiative effect | SWCF |
| Total cloud amount | CLDTOT |
| 200 hPa zonal wind | U |
| 500 hPa geopotential height | Z3 |
| Precipitation rate | PRECT |
| Total precipitable water | TMQ |
| Sea level pressure | PSL |
| Surface latent heat flux | LHFLX |
| Surface sensible heat flux | SHFLX |
| Surface stress | TAUX, TAUY |
| 2m air temperature | TREFHT |
| Sea level temperature on land | TS |

# Appendix C: Additional figures

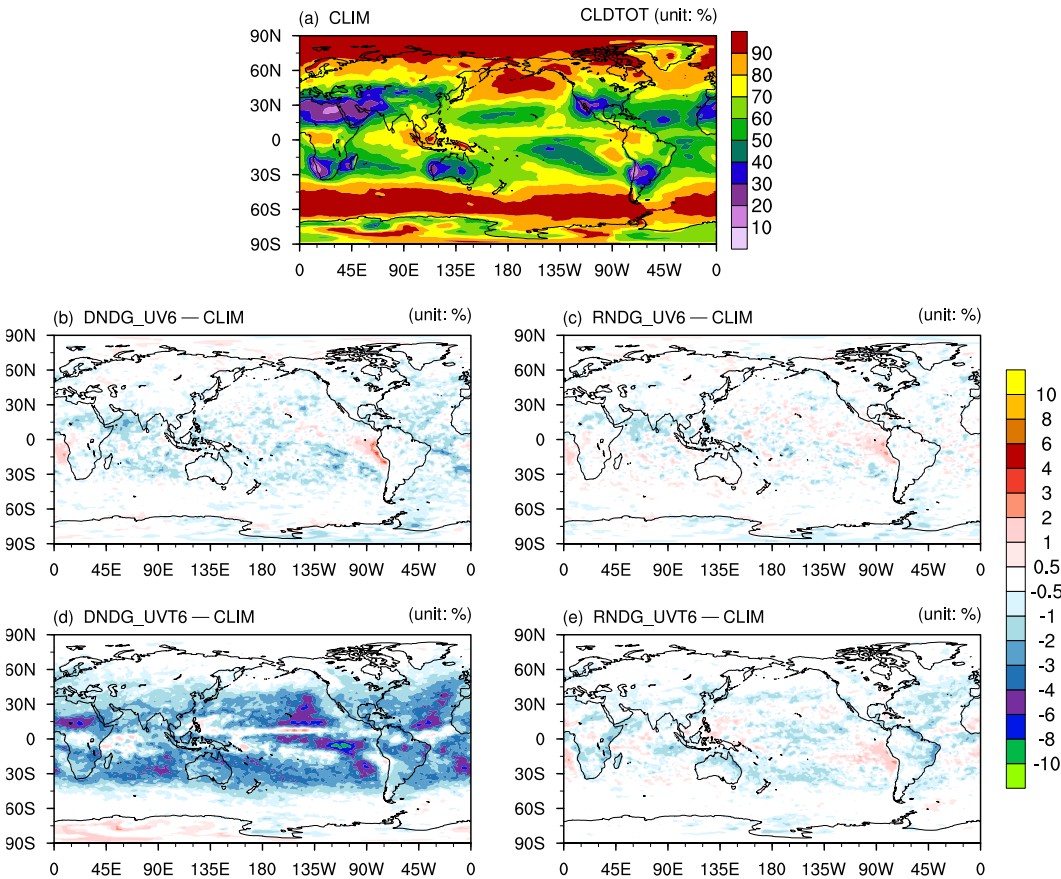

**Figure C1.** As in Figure 2 but showing results for the total cloud fraction (CLDTOT, unit: percent). The simulation setups are described in Section 2.3 and Table 1.

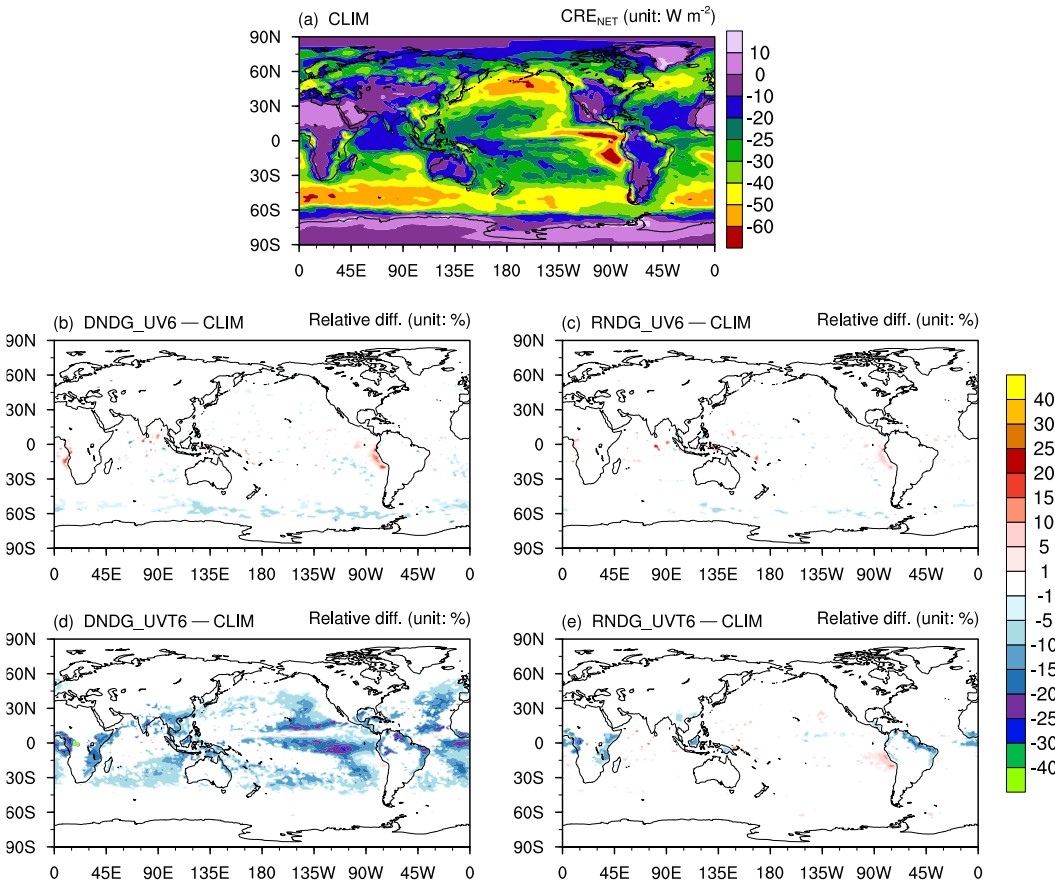

**Figure C2.** As in Fig. 2 but showing the relative differences of $\mathrm{CRE_{NET}}$ (unit: %) in panels (b)–(e) in contrast to the non-normalized differences shown in Fig. 2b-e. The simulation setups are described in Section 2.3 and Table 1.

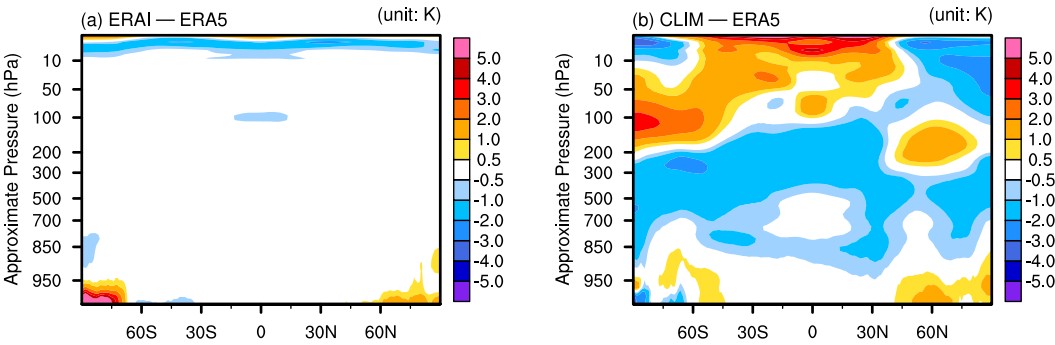

**Figure C3.** Year 2010 annual mean zonally averaged differences in temperature (T, unit: K) between ERA-Interim ("ERAI") and ERA5 (panel a), and between EAMv1's free-running simulation CLIM and ERA5 (panel b).

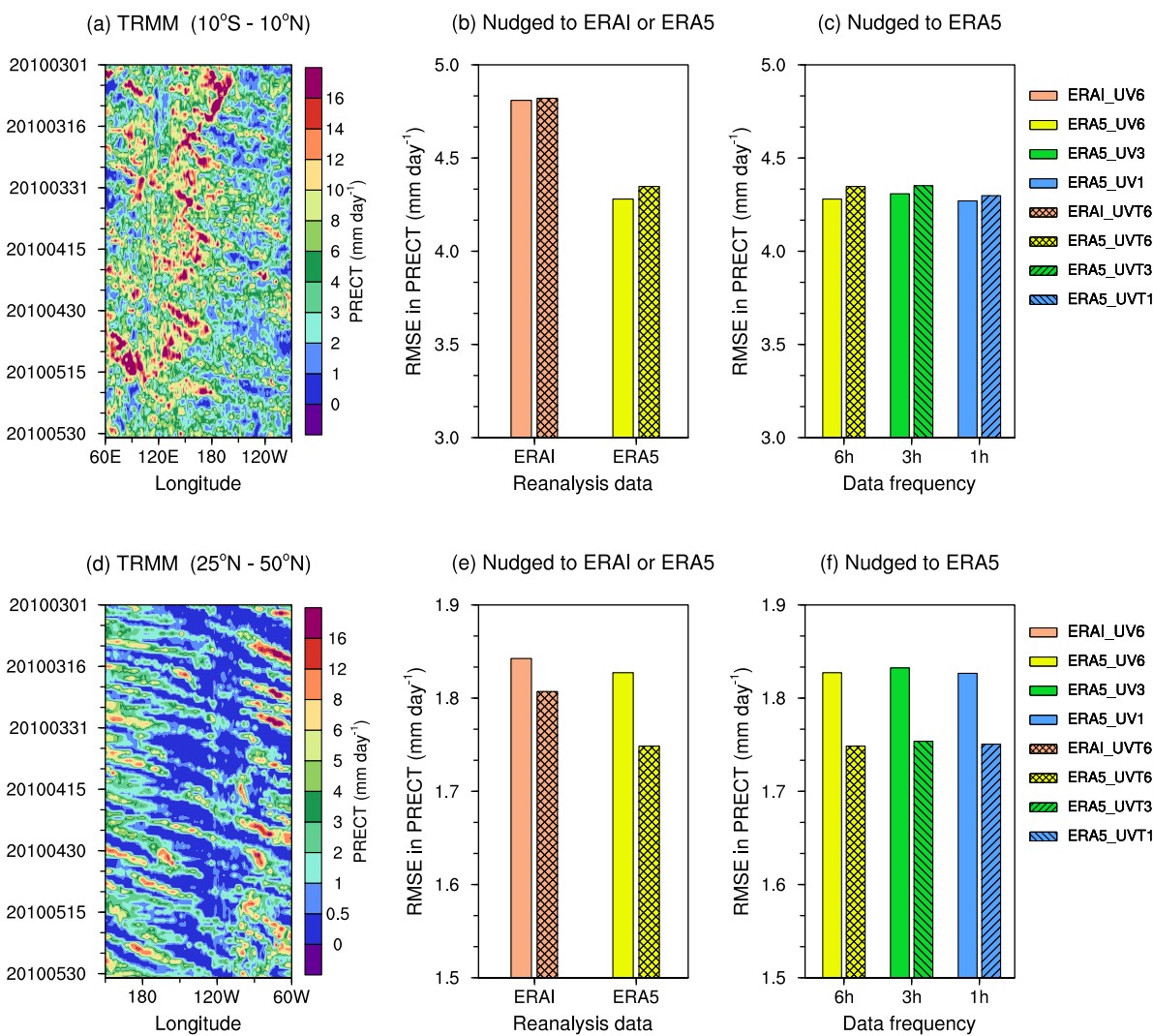

**Figure C4.** As in Fig. 9, but the bar charts in panels (b–c) and (e–f) show the root-mean-square differences (instead of correlations) between the Hovmöller diagrams derived from the various nudged simulations (not shown) and the Hovmöller diagrams derived from TRMM (shown as panels a and d). The simulation setups are described in Section 2.3 and Table 1.

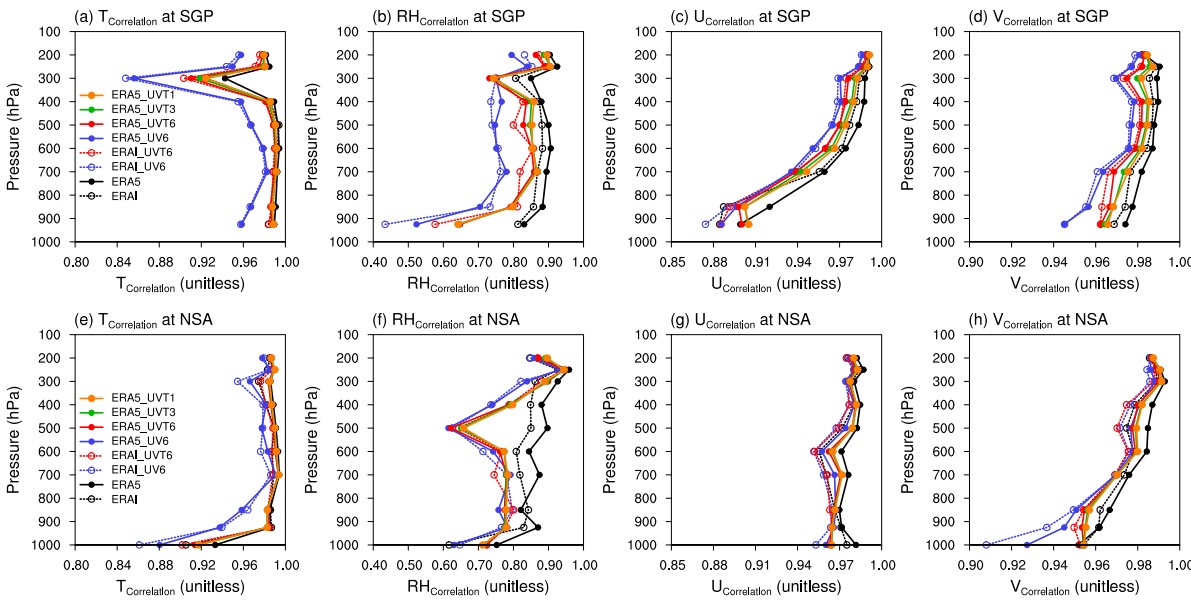

**Figure C5.** As in Fig. 10 but showing the temporal correlations (instead of RMSEs) between various nudged simulations (colored lines) or reanalysis products (ERA-Interim and ERA5, black lines) and ARM measurements from SGP (upper row) and NSA (lower row). The simulation setups are described in Section 2.3 and Table 1.

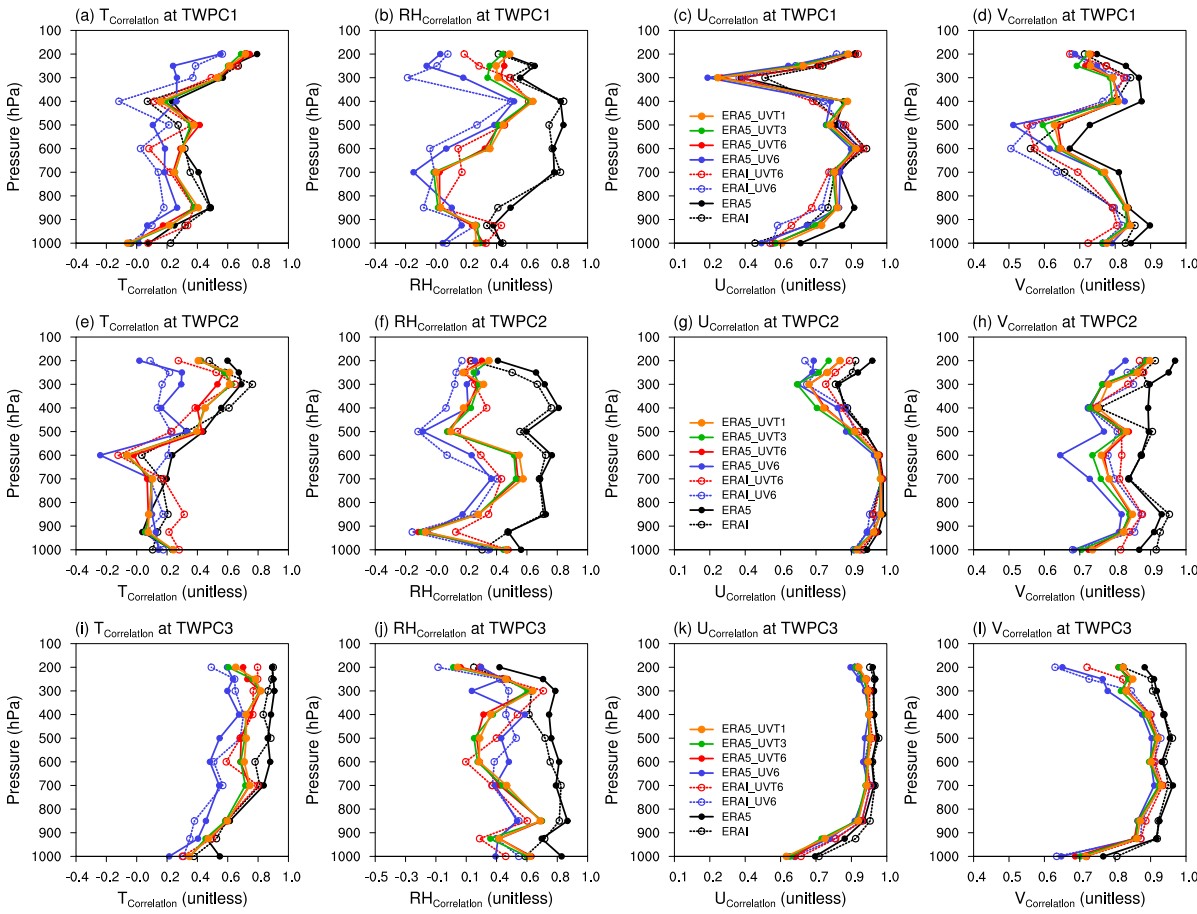

**Figure C6.** As in Fig. 11 but showing the temporal correlations (instead of RMSEs) between various nudged simulations (colored lines) or reanalysis products (ERA-Interim and ERA5, black lines) and ARM measurements from the three central facilities at the TWP site (TWPC1–C3, top to bottom). The simulation setups are described in Section 2.3 and Table 1.

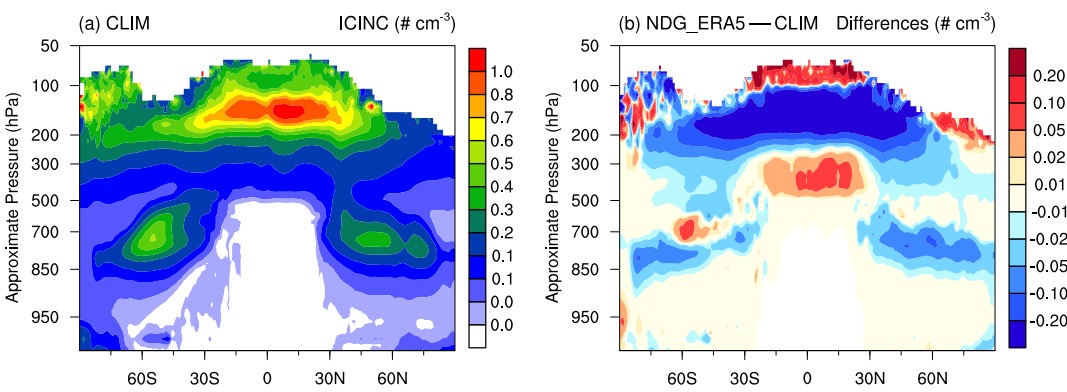

**Figure C7.** Annual mean zonally averaged (a) in-cloud ice number concentration (ICINC, unit: # cm$^{-3}$) from the free-running simulation CLIM and (b) difference in ICINC (unit: # cm$^{-3}$) between CLIM and a nudged simulation. "NDG_ERA5" in the title of panel (b) refers to the simulation RNDG_ERA5_UVT3 in which both the horizontal winds and temperature are nudged towards 3-hourly data from the ERA5 reanalysis). All simulations used present-day (PD) aerosol emissions. See details in Section 2.3 and Table 1.

*Code availability.* The EAMv1 source code and run scripts used in this study can be found on Zenodo at https://doi.org/10.5281/zenodo.
5532606 (E3SM developers et al., 2021).

*Data availability.* The data used by figures in the paper can be found on Zenodo at https://doi.org/10.5281/zenodo.6988262 (Zhang and
Zhang, 2022). This includes (1) post-processed EAMv1 output, (2) reanalysis and satellite data interpolated to the same grid as in (1), and (3)
radiosonde measurements at selected sites. The radiosonde measurements were obtained from ARM (http://dx.doi.org/10.5439/1021460), a
U.S. Department of Energy Office of Science user facility managed by the Biological and Environmental Research program. Satellite data
used in this paper were obtained from the following sources

– https://pmm.nasa.gov/data-access/downloads/trmm for the TRMM 3B42 precipitation data,

– https://climatedataguide.ucar.edu/climate-data/persiann-cdr-precipitation-estimation-remotely-sensed-information-using-artificial for
the PERSIANN-CDR precipitation data,

– https://climatedataguide.ucar.edu/climate-data/outgoing-longwave-radiation-olr-hirs for the HIRS OLR data,

– https://www.esrl.noaa.gov/psd/data/gridded/data.interp_OLR.html for the AVHRR OLR data.

*Author contributions.* KZ and HW initiated this study. HW identified the two issues in the implementation of nudging in the default EAMv1.
SZ conducted all simulations, processed the model output, and carried out the analyses with input from KZ and HW. SZ, KZ, and HW wrote
the paper. All co-authors contributed to the revisions.

*Competing interests.* The authors declare no competing interests

*Acknowledgements.* The authors thank Dr. William Collins and the anonymous referee for their careful review and helpful comments. This
work was supported primarily by the Laboratory Directed Research and Development Program at Pacific Northwest National Laboratory
(PNNL), a multiprogram national laboratory operated by Battelle for the U.S. Department of Energy (DOE) under contract DE-AC05-
76RL01830. KZ was partially supported by DOE Office of Science, Biological and Environmental Research (BER) via the Energy Exascale
Earth System Model (E3SM) project. HW was supported by DOE's Scientific Discovery through Advanced Computing (SciDAC) program
via a partnership in Earth system model development between BER and Advanced Scientific Computing Research (ASCR). The EAMv1 code
(DOI:10.11578/E3SM/dc.20180418.36) was obtained from the E3SM project sponsored by DOE BER. Computing resources were provided
by DOE BER Earth System Modeling program's Compy computer cluster located at PNNL. The ARM data used for model evaluation in
this study was obtained from ARM (https://arm.gov/), a DOE Office of Science user facility managed by BER.

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
