# Peer review of "Further improvement and evaluation of nudging in the E3SM Atmosphere Model version 1 (EAMv1): simulations of the mean climate, weather events, and anthropogenic aerosol effects"

_Geoscientific Model Development, 2022_

## Author Comment (AC1)

**Point-by-point Reply to Comments from Reviewer 1**

We greatly appreciate the reviewer's comments and suggestions. Below please find our reply (dark red).

Zhang et al. (2022) improved the EAMv1 by revising the sequence of nudging tendency in the model and increasing the nudging frequency. The manuscript has thoroughly discussed the nudging impacts from several sets of model simulations and provided suggestions for the nudged simulations. The manuscript is well organized and has addressed key issues in the nudged model simulations in EAMv1. Below are a few comments.

**General comment**

The EAMv1 nudging simulations are improved in this work. My major concern is the model resolution used in this work. The current model is configured to be 1 deg. It does not fully take advantage of reanalysis data with higher spatial resolution, which may dilute the impacts from ERAI and ERA5. On the other hand, if the model is configured to a higher spatial resolution (which is the direction in the global climate community), the model accuracy to simulation important small-scale processes could also be improved. In that case, it may require a different nudging strength or frequency.

For the current study, we focused on the analysis using the 1-deg model, because it's the standard resolution for v1 and v2 of E3SM (Golaz et al., 2019, Golaz et al., 2022) and is used for most model applications. But we agree with the reviewer that it would be useful to test the nudging implementation (e.g., relaxation time scale, frequency) at higher spatial resolutions (especially for 25km, which is close to the ERA5 resolution), so that we could maximize the benefit of using the high-resolution reanalysis data and improve the global atmospheric hindcast simulations.

We have made clarifications in the abstract and introduction to explicitly state that this study focuses on the E3SMv1 model that is configured with a 1-deg horizontal resolution.

References:

Golaz, J.-C., Caldwell, P. M., Van Roekel, L. P., Petersen, M. R., Tang, Q., Wolfe, J. D., et al. (2019). The DOE E3SM coupled model version 1: Overview and evaluation at standard resolution. Journal of Advances in Modeling Earth Systems, 11, 2089– 2129. https://doi.org/10.1029/2018MS001603

Golaz, J.-C., Van Roekel, Luke P., et al. (2022) The DOE E3SM Model Version 2: Overview of the physical model, Earth and Space Science Open Archive, doi: 10.1002/essoar.10511174.1. https://www.essoar.org/doi/10.1002/essoar.10511174.1

**Specific comments:**

Page 9, Figure 3, typo? No magenta box in Figure 4e. Maybe Figure 2e?

Yes. Corrected.

Page 21, line 350-351, there is no Table S3 in the Supplement.

There might be some technical issue. Table S3 in the original Supplement (https://gmd.copernicus.org/preprints/gmd-2022-10/gmd-2022-10-supplement.pdf) does show up on our computer. We will double-check it for the revised submission.

Also, I'm confused about the sign between line 351 and in Figure 12. Suggest to clarify the discussion here.

Figure 12 illustrates the ratio of the values derived from the nudged simulation to the values derived from the CLIM simulation. As the signs of aerosol-induced effects from the nudged simulation and CLIM are in the same sign, we will always have a positive ratio. We have made further clarifications in Figure 12 and Section 5 of the revised manuscript.

Page 21, line 365-369, this is also a typical approach in many other earth system models.

Yes, we agree. We slightly adjusted the wording in Section 5 of the revised manuscript and cited the previous studies on many other earth system models.

---

## Author Comment (AC2)

**Point-by-point Reply to Comments from Reviewer 2**

We greatly appreciate the comments from Dr. William Collins. Below please find our reply (dark red).

This was a clear study that successfully explained some biases in the previous nudging schemes. The comments on time resolution will be useful to the community which generally used 6-hourly data.

For the aerosol forcing why is 6-hour nudging used when the authors have shown that 3-hour is preferred? This experiment should be repeated with the 3-hour resolution.

We have performed the nudged simulations with the 3-hour constraining data and evaluated the impact on the effective aerosol forcing (Faer) estimate. As shown in the following figure, the global annual mean Faer estimated by the nudged simulations is not sensitive to the choice of the constraining data frequency.

For both UV-nudging (Figure 1a-b) and UVT-nudging (Figure 1c-d), the Faer estimates are similar for simulations with 6h constraining data (orange) and for those with 3-hour constraining data (blue). This applies to both the simulations nudged towards the model's own meteorology (solid) and simulations nudged towards ERA5 (hatched). The impact of constraining data frequency on present-day simulations is relatively small except in regions where strong diurnal variations exist (Section 3.2). Therefore, the impact on global and tropical mean estimates is small. If the investigation is for regions with strong diurnal cycles, using the 3h constraining data should be a better solution. We have added these discussions in Section 5 in the revised manuscript.

**Figure 1.** Global (a, c) and topical (b, d) mean annually averaged anthropogenic aerosol radiation effect (PD-PI differences, denoted by  $\Delta$ ) estimated by free-running (i.e., CLIM, grey bars) and nudged EAM simulations (colored bars). Shown in each panel is the ratio of the values from each simulation to corresponding values in CLIM. The original data can be found in Table S3. The vertical thin line and whiskers on the dark grey bar indicate the standard deviation of the 5-ensembles in CLIM. FSNT/FLNT is the TOA net shortwave/longwave radiation flux and SWCF/LWCF the shortwave/longwave cloud radiative effects. The top row compares the UV-nudging simulations with CLIM, while the bottom row compares the UVT-nudging simulations with CLIM. The solid color bars are for the nudged to CLIM simulations, while the hatched color bars are for the nudged to ERA5 simulations. The simulations are described in Section 2.3 and Table 1.

The authors could explain the issues with the temperature nudging more fully. The effects are attributed to biases. While this could be true for ERA, this should be much less significant for CLIM (fig 2(e)) and would be even less so for 3-hourly (in fig 4(d)). Is this actually removing a meteorological adjustment to the aerosols?

This is a good question. Our original statement is for the simulations nudged towards reanalysis. We have made additional analyses to further explain this and the effect in simulations nudged towards CLIM.

For the simulations (PD and PI) nudged towards CLIM (PD), we think the significant impact of temperature nudging is indeed due to the more constrained meteorological adjustment to the aerosol perturbation. The anthropogenic aerosols do have a small but significant impact on temperature, even when horizontal winds are nudged (c.f. Figure 2a–c). When the temperature is nudged, the associated responses in cloud processes could be changed (c.f. Figure 2d–f for cloud liquid water mixing ratio and Figure 2g–i for cloud ice water mixing ratio).

---

## Author Response (AR1)

**Point-by-point response to the reviews**

**Shixuan Zhang and Kai Zhang (on behalf all authors)**

**June 2022**

We greatly appreciate the careful reviews and helpful comments from the anonymous reviewer and Dr. William Collins. Based on their comments and suggestions, we have made the following changes in the revised manuscript:

- 1. The spatial resolution of E3SMv1 used for this study has been clarified in the abstract, conclusion, and other relevant places.
- 2. Additional information about the nudging implementation in E3SMv1 is added to Section 2.2.
- 3. A detailed description of the differences in emissions between pre-industrial (PI, year 1850) and present-day (PD, year 2010) simulations is added to Section 2.3 and the caption of Table S3 in the supplement.
- 4. Section 5 "Impact on the estimation of anthropogenic aerosol effect", has been revised and reorganized. New figures (12-14, A7, and A8) have been added to discuss: a) the impact of constraining data frequency on the effective aerosol forcing (Faer) estimate, and b) the impact of the temperature nudging on the Faer estimate in the nudged EAM simulations.
- 5. Typo corrections and writing improvement have been made throughout the manuscript.

The point-by-point responses to the reviewer's comments are presented below.

**Reply to Referee 1**

**Referee comment:** The EAMv1 nudging simulations are improved in this work. My major concern is the model resolution used in this work. The current model is configured to be 1 deg. It does not fully take advantage of reanalysis data with higher spatial resolution, which may dilute the impacts from ERAI and ERA5. On the other hand, if the model is configured to a higher spatial resolution (which is the direction in the global climate community), the model accuracy to simulation important small-scale processes could also be improved. In that case, it may require a different nudging strength or frequency.

**Author response:** For the current study, we focused on the analysis using the 1-deg model, because it's the standard resolution for v1 and v2 of E3SM (Golaz et al., 2019, Golaz et al., 2022) and is used for most model applications. But we agree with the reviewer that it would be useful to test the nudging implementation (e.g., relaxation time scale, frequency) at higher spatial resolutions (especially for 25km, which is close to the ERA5 resolution), so that we could maximize the benefit of using the high-resolution reanalysis data and improve the global atmospheric hindcast simulations.

We have made clarifications in the abstract, conclusion, and other relevant places to explicitly state that this study focuses on the E3SMv1 model that is configured with a 1-deg horizontal resolution.

Referee comment: Page 9, Figure 3, typo? No magenta box in Figure 4e. Maybe Figure 2e?

Author response: The correction has been made in the revised manuscript.

Referee comment: Page 21, line 350-351, there is no Table S3 in the Supplement.

**Author response:** There might be some technical issue. Table S3 in the original Supplement (https://gmd.copernicus.org/preprints/gmd-2022-10/gmd-2022-10-supplement.pdf) does show up on our computer. We have double-checked the Tables in Supplement in the revised submission.

**Referee comment:** Also, I'm confused about the sign between line 351 and in Figure 12. Suggest to clarify the discussion here.

**Author response:** Sorry for the confusion. Figure 12 illustrates the ratio of the values derived from the nudged simulation to the values derived from the CLIM simulation. As the signs of aerosol-induced effects from the nudged simulation and CLIM are the same, the ratio will always be positive. We have made revisions in Figure 12 and clarifications in Section 5 of the revised manuscript.

**Referee comment:** line 365-369, this is also a typical approach in many other earth system models.

**Author response:** We agree with the reviewer. This is addressed by adjusting the wording in Section 5 of the revised manuscript and cited the previous studies on many other earth system models:

"Overall, consistent with previous studies using other global aerosol-climate models (e.g. Kooperman et al., 2012; Zhang et al., 2014; Ghan et al., 2016), our results indicate that nudging the horizontal winds but not temperature towards the ERA5 reanalysis or EAM's own meteorology is the preferred simulation configuration to estimate Faer. The temperature nudging needs to be applied with caution, as the potential climatology discrepancies between CLIM and reanalysis might lead to large biases in the Faer estimation."

**References:**

Kooperman, G. J., Pritchard, M. S., Ghan, S. J., Wang, M., Somerville, R. C. J., and Russell, L. M. (2012), Constraining the influence of natural variability to improve estimates of global aerosol indirect effects in a nudged version of the Community Atmosphere Model 5, J. Geophys. Res., 117, D23204, doi:10.1029/2012JD018588.

Zhang, K., Wan, H., Liu, X., Ghan, S. J., Kooperman, G. J., Ma, P.-L., Rasch, P. J., Neubauer, D., and Lohmann, U.: Technical Note: On the use of nudging for aerosol–climate model intercomparison studies, Atmos. Chem. Phys., 14, 8631–8645, https://doi.org/10.5194/acp-14-8631-2014, 2014.

Ghan, S., Wang, M., Zhang, S., Ferrachat, S., Gettelman, A., Griesfeller, J., Kipling, Z., Lohmann, U., Morrison, H., Neubauer, D., Partridge, D. G., Stier, P., Takemura, T., Wang, H., and Zhang, K.: Challenges in constraining anthropogenic aerosol effects on cloud radiative forcing using present-day spatiotemporal variability, Proceedings of the National Academy of Sciences, 113, 5804–5811, https://doi.org/10.1073/pnas.1514036113, 2016.

**Reply to Referee 2**

**Referee comment:** This was a clear study that successfully explained some biases in the previous nudging schemes. The comments on time resolution will be useful to the community which generally used 6-hourly data. For the aerosol forcing why is 6-hour nudging used when the authors have shown that 3-hour is preferred? This experiment should be repeated with the 3-hour resolution.

Author response: Thanks for the positive feedback and comments. Following the reviewer's suggestion, we have performed the nudged simulations with the 3-hourly nudging data and discussed the impact on the effective aerosol forcing ( $F_{acr}$ ) estimate in Section 5:

In Figure 14, we evaluate the impact of the frequency of the constraining data. At least for the global and annual mean  $F_{aer}$ , the results obtained from simulations using 6-hourly constraining data (orange bars in the figure) are very similar to those obtained using 3-hourly constraining data (blue bars), regardless of whether UV-nudging (Fig. 14, upper row) or UVT-nudging (Fig. 14, lower row) is used. The small impact of constraining data frequency on global and tropical mean  $F_{aer}$  estimates is expected. As shown in Section 3.2, the impact of constraining data frequency on present-day simulations is sizable only in limited regions where strong diurnal variations exist. Therefore, using 6-hourly constraining data in nudged simulations is sufficient for estimating the time-mean  $F_{aer}$ .

**Figure 14.** Global mean (a, c) and topical mean (b, d) annually-averaged anthropogenic aerosol effect (PD-PI differences, denoted by  $\Delta$ ) estimated by free-running (i.e. CLIM, grey bars) and nudged EAM simulations (colored bars). FSNT and FLNT are the TOA net shortwave and longwave radiation flux, respectively. SWCF and LWCF are the shortwave and longwave cloud radiative forcing, respectively. All values have been normalized by the ensemble

mean of CLIM. The thick whiskers attached to the grey bars indicate the two-standard-deviation ranges of the 5member CLIM ensemble. The upper row compares the UV-nudged simulations with CLIM, and the lower row compares the UVT-nudged simulations with CLIM. The solid color bars indicate simulations nudged towards CLIM; the hatched color bars indicate simulations nudged towards ERA5 reanalysis. Orange and blue bars correspond to nudged simulations performed with 6-hourly and 3-hourly constraining data, respectively. The simulations are described in Section 2.3 and Table 1.

**Referee comment:** The authors could explain the issues with the temperature nudging more fully. The effects are attributed to biases. While this could be true for ERA, this should be much less significant for CLIM (fig 2(e)) and would be even less so for 3-hourly (in fig 4(d)). Is this actually removing a meteorological adjustment to the aerosols?

Author response: This is a good question. Our original statement is for the simulations nudged towards reanalysis. We do think the additional temperature nudging will constrain the meteorological adjustment to the aerosol perturbation, which will lead to a biased  $F_{aer}$  estimate even for the simulations nudged towards CLIM. We have added a new figure (A8) in the appendix and discussed it in Section 5:

Figure 12a–b indicates that when EAMv1 simulations are nudged to its own climatology, constraining temperature also has significant impacts on the estimated  $\Delta$ FSNT and  $\Delta$ SWCF (see green versus pink bars with solid fill). This is mainly due to the constrained temperature adjustment to the aerosol perturbation, since the PD and PI simulations were nudged towards the same CLIM PD simulation. The anthropogenic aerosols and precursors are known to have significant impacts on air temperature (Fig. A8a). When only the horizontal winds are nudged towards CLIM PD, the impacts of anthropogenic aerosols and precursors on temperature are smaller than in the free-running simulations but nevertheless still sizable (Fig. A8b). In contrast, the nudging of temperature substantially reduces the PD-PI temperature differences as expected (Fig. A8c). The results shown in Fig. A8 suggest that the constrained temperature response mainly affects the simulated PD-PI changes in cloud liquid mass ( $\Delta$ CLDLIQ, Fig. A8, second row) and cloud ice mass ( $\Delta$ CLDICE, Fig. A8, third row) in the middle and lower troposphere (i.e., below 500hPa). This explains why the solid green bars in Fig. 12 deviate from the gray bars more in the shortwave radiation than in the longwave component.

---

## Author Response (AR2)

**Point-by-point Response to Reviewer's Comments**

Kai Zhang, Hui Wan, and Shixuan Zhang

August 2022

We would like to express our sincere thanks to Dr. William Collins for his very insightful and constructive comments and suggestions. His second review helped us to further improve the manuscript substantially. Our detailed responses can be found below in black. The reviewer's comments are shown in blue.

In addition, we would like to ask for the Topical Editor's approval for adding a few words to the title of the manuscript to reveal what aspects of the EAM simulations are analyzed in this study.

- Old title: *Further improvement and evaluation of nudging in the E3SM Atmosphere Model version 1 (EAMv1)*

- Revised title: *Further improvement and evaluation of nudging in the E3SM Atmosphere Model version 1 (EAMv1): simulations of the mean climate, weather events, and anthropogenic aerosol effects*

**Responses to reviewer's comments**

**Comment 1:**

I wish to thank the authors for their careful consideration of my comments, particularly to my question on temperature nudging. Their response and the text lines 400-430 shows that there are two effects – a mean-state correction in ERA5, and a suppression of adjustments in CLIM PD. It would be helpful to explicitly explain it in these terms.

Thanks for the positive feedback and the nice summary of the issues associated with temperature nudging. Following the reviewer's suggestion, we have significantly revised the abstract, Section 5, and the conclusions section:

- The 3$^{rd}$ paragraph of the abstract has been rewritten.
- Section 5 ("Estimation of the anthropogenic aerosol effects") has been significantly rewritten. In addition, we added the following subsection titles to help guide the readers through our reasoning and make our point:
    - Section 5.1 Results from the free-running EAMv1
    - Section 5.2 Impacts of temperature nudging

- The second last paragraph in the conclusions section, which summarized our results on the estimation of the aerosol effects, has also been rewritten.

Please see the revised manuscript or the file with tracked changes for the details of our revision.

**Comment 2:**

The mean-state correction is presumably a more realistic calculation of the ice-cloud forcing so it is not obvious that ignoring this correction should be the preferred calculation (which line 433 states). Line 22 refers to the temperature-nudged calculation as "a slightly biased estimate". However figure 13 shows that the ERA5 temperature nudging actually removes a systematic bias, so it could be argued that this is "a less biased estimate". The problem is that ERA5 temperature nudging also removes the temperature adjustment.

We agree that whether an estimate is biased depends on what the reference is and what our goal is, so the phrase "biased estimate" in our earlier version lacked clarity. In the revised manuscript, we clarify that our preferred configurations of nudging are those capable of producing estimates of anthropogenic aerosol effects that agree with the results from the free-running EAM simulations. For this reason, the mean bias correction caused by nudging temperature to ERA5 and the resulting changes in the anthropogenic aerosol effects are undesirable for our intended applications of nudging. We have clarified this perspective in the abstract, Section 5, and the conclusions section in the revised manuscript. Also, we no longer use the phrase "biased estimate" but use instead "differences" or "discrepancies" with respect to the free-running simulations.

Regarding the comment "The problem is that ERA5 temperature nudging also removes the temperature adjustment", we clarify in the revised manuscript that ERA5 temperature nudging has both effects: mean bias correction and removal of the temperature response to aerosol forcing.

**Comment 3:**

The calculation of the temperature-mediated adjustment is +24% in FSNT (i.e. from -1.862 to -2.453 W/m2 in table 3), but smaller (+8%) in the net. This is a useful scientific point that helps our understanding of the overall radiative effects of aerosols and so should be mentioned in section 5 and also in the conclusions, for instance the adjustment would be absent from an offline calculation and from the classical split into "direct" and "albedo" effects.

Thanks for pointing this out. We have added the following statements to Section 5 after describing the impact of temperature nudging on the shortwave and longwave components of the TOA fluxes and the cloud radiative effects in Section 5.2.3:

*"It is worth noting that the discrepancies in $\Delta F_{NET}$ might appear to be not as large. For example, we see a 15% difference in the global mean in (Fig. 12a and Table S4) and a 17% difference in the tropical average (Fig. 12b and Table S5). But the smaller differences in the net fluxes are results of the cancellation of large changes in the shortwave and longwave components caused by temperature nudging."*

The following statement has been added to the second last paragraph of the conclusions:

*"While the percentage discrepancies in the net TOA flux and CRE appeared to be considerably smaller, this was the result of the cancellation of large discrepancies in the shortwave and longwave components."*

**Comment 4:**

I suggest lines 431-435 are rewritten to explain that nudging to ERA5 might remove a mean-state bias, but because it also supresses a temperature-mediated adjustment it is not the preferred estimate. Similar wording changes are needed in the abstract lines 20-22.

This has been addressed during the rewrite of Section 5 and the 3rd paragraph of Section 5.

**Comment 5:**

Line 125: This needs a short explanation to say that Faer is the Effective Radiative Forcing including all meteorological adjustments (citing for instance AR5 or AR6) defined using fixed-SSTs. You can then refer back to this explanation when you find suppression of adjustments in section 5.

Thanks for the helpful suggestion. We have added a new paragraph to Section 2.3 ("Simulation"):

*"The anthropogenic aerosol effects we are interested in estimating in this study is the effective radiative forcing (ERF) defined in the Fifth Assessment Report of the Intergovernmental Panel on Climate Change, namely the changes in the TOA radiative fluxes when all physical variables in a climate model are allowed to respond to perturbations except for those concerning the ocean and sea ice (Myhre et al., 2014). Our primary focus is the net TOA flux and its shortwave and longwave components. These are denoted by $F_{NET}$, $F_{SW}$ and $F_{LW}$, respectively, in the remainder of the paper, with positive values indicating fluxes downward (i.e., into the atmosphere). For the readers who have worked with EAM's output, we note that the $F_{SW}$ presented here is EAM's output variable FSNT while the $F_{LW}$ here is $-FLNT$, as FLNT in EAM is defined to be positive upward. The net flux is calculated as*

$$F_{NET} = F_{SW} + F_{LW} = FSNT - FLNT$$

*The changes in $F_{NET}$, $F_{SW}$ and $F_{LW}$, caused by anthropogenic aerosols are denoted by $\Delta F_{NET}$, $\Delta F_{SW}$ and $\Delta F_{LW}$, respectively."*

In Section 5 of the revised manuscript, we clarify again that the aerosol effects we try to estimate are the Effective Radiative Forcing (ERF).

**Comment 6:**

Line 173: Please explain how CF is calculated. Is it a double-call model diagnostic with clear and cloudy-sky, or is it an offline calculation?

To help clarify this, we have added a second new paragraph to Section 2.3 ("Simulation"):

*"We are also interested in the impact of anthropogenic aerosols on the cloud radiative effect (CRE). CRE is defined as the change in a TOA radiative flux caused by the presence of clouds; here we denote the CRE on the net, shortwave and longwave TOA radiative flux by $CRE_{NET}$, $CRE_{SW}$ and $CRE_{LW}$, respectively, with positive values indicating more fluxes into the atmosphere. The $CRE_{SW}$ and $CRE_{LW}$ presented here are EAM's output variables SWCF and LWCF, respectively, both of which are diagnosed during a simulation by performing the radiation calculations twice (with and without clouds) and then computing the difference. The net CRE is calculated by*

$$CRE_{NET} = CRE_{SW} + CRE_{LW} = SWCF + LWCF$$

*The changes in $CRE_{NET}$, $CRE_{SW}$ and $CRE_{LW}$ caused by anthropogenic aerosols are denoted by $\Delta CRE_{NET}$, $\Delta CRE_{SW}$ and $\Delta CRE_{LW}$, respectively."*

Please also note that we have changed our wording from Cloud Forcing (CF) to Cloud Radiative Effect (CRE) to be consistent with most recent IPCC reports.

**Comment 7:**

Line 388 and table S3. The sign convention for the LW forcing should be explained. I assume it is downward. It might be clearer to use the same direction (outgoing) for both SW and LW.

Sorry about the confusion. Some of the EAM output variables define downward fluxes to be positive and some other fluxes are positive upward. To help improve clarity,
- We introduced two sets of new notations (for the TOA fluxes and CRE, respectively), to use consistent sign convention and to distinguish from EAM's output variables. These new notations are introduced in the two new paragraphs quoted in our replies to comments 5 and 6 above and are used throughout the manuscript.

- We added a new table S1 to the Supplementary Materials to explain the meaning and sign convention of the EAM output variables shown in the tables in the Supplementary Materials.

Hope these are helpful for avoiding further confusion.